# Reliability and validity of the Hungarian version of the Personality Inventory for DSM-5 (PID-5)

Zita S. Nagy[1], Ella Salgó[2], Bettina Bajzát[2], Bálint Hajduska-Dér[2], Zsolt Szabolcs Unoka[2]*

1 National Institute for Medical Rehabilitation, Budapest, Hungary, 2 Department of Psychiatry and Psychotherapy, Semmelweis University, Budapest, Hungary

* unoka.zsolt@med.semmelweis-univ.hu

## Abstract

### Objectives and methods

In order to assess the internal consistency, fit indexes, test-retest reliability, and validity of the Personality Inventory for the DSM-5 (PID-5) and its associations with age, gender, and education, 471 non-clinical (69,6% female; mean age: 37,63) and 314 clinical participants (69,7% female, mean age: 37,41) were administered the Hungarian translation of the PID-5, as well as the SCL-90-R and the SCID-II Personality Questionnaire.

### Results

We found that; (a) temporal consistency of the Hungarian PID-5 was confirmed by one-month test-retest reliability analysis, (b) validity of the PID-5 instrument is acceptable in the clinical and the non-clinical sample as well, based on significant correlations with SCID-II and SCL-90-R, (c) PID-5 facets' and domains' associations with gender, age, and level of education are in accordance with previous findings.

### Conclusion

These findings support that the Hungarian PID-5 is a reliable and valid instrument for both clinical and non-clinical populations.

## 1. Introduction

In the Alternative Model for Personality Disorders (AMPD) in the Diagnostic and Statistical Manual of Mental Disorders (DSM-5, section III), specific personality disorders (PD) are defined by typical impairments in personality functioning (Criterion A) and characteristic pathological personality traits (Criterion B) [1]. The AMDP retained only 6 of the 10 DSM-IV PDs and changed the Personality Disorder Not Otherwise Specified (PDNOS) category to a PD-Trait Specified diagnosis. The AMDP has not been accepted as the official diagnostic

**Data Availability Statement:** All data files are available at OSF: https://osf.io/zcm92/.

**Funding:** This work was supported by the following grants: Hungarian National Research, Development and Innovation Fund [grant numbers NKFI-

132546]. Zsolt Szabolcs Unoka is the PI. Hungarian National Research, Development and Innovation Fund [grant numbers NKFI-129195]. Zsolt Szabolcs Unoka is a co-investigator.

**Competing interests:** The authors have declared that no competing interests exist.

system for use in the DSM-5, partly because of the limited research support for its specific components. Since its publication, more than hundred publications investigated the validity of the AMDP in different languages [3, 7] and it needs further validation and examination whether it is generalizable to Hungarian language and culture. Although another study has already been published about the psychometric properties of the PID-5 in a Hungarian clinical and community sample [19], we found that this study had some psychometric and scientific limitations besides its strengths, and it is worth using another adaptation of the PID-5 mesure. First of all, our Hungarian translation of the PID-5 scale had been approved by the American Psychiatric Association (APA).

Moreover, we added some important extensions to the investigation of the psychometric properties of the PID-5 by including a 1-month test-retest reliability of the scales, by using SCID-II and SCL-90-R correlations as validity criteria, and by presenting three different models of the PID-5 internal structure in the confirmatory factor analysis.

Pathological personality traits (Criterion B), proposed by Krueger, Derringer, Markon, Watson, and Skodol [2] are organized into five broad domains: Negative Affectivity, Detachment, Antagonism, Disinhibition, and Psychoticism. Within the five broad trait domains, there are 25 specific trait facets [1]. APA (https://www.psychiatry.org/psychiatrists/practice/dsm/educational-resources/assessment-measures) provided different kinds of measures (self-report, informant, rating scale) for assessing maladaptive personality traits. The Personality Inventory is a self-report and other-informant measure for the DSM–5 (PID–5) [2]. Since its introduction, several studies examined various aspects of PID-5's psychometric properties in non-clinical and clinical samples (see a review by Al-Dajani et al. [3]). We aim to further examine the psychometric properties of the Hungarian version of PID-5 in non-clinical and clinical samples. Our decision to use a non-clinical sample as well is based on the theory that traits that describe personality disorders are experienced at a less extreme level among healthy persons [4].

Internal consistencies of PID-5 domains and facets have been studied extensively, and mixed internal consistency scores were demonstrated. Coefficient alphas [5] for domain scales were between .75-.92 and .87-.96 and for facets ranged from .46-.77 to .94-.96, demonstrating some possible issues with scale reliability at the facet level (see Al-Dajani et al. [3]). Two versions of the domain scoring algorithm are used in the literature. The 25 facets version, which uses all the 25 facets in the definitions of the five domains, and the 15 facets version, which uses only 3 facets per domain [6]. Both versions derived from the study of Krueger et al. [2] which identified five higher order factors by explorative factor analysis of 25 facets. In order to eliminate cross-loadings of facets, three traits with the highest factor loadings were selected for each domain [2]. This scoring instruction was published in the APA copyright version of the PID-5 [6]. Further studies that investigated the five-factor exploratory factor model of the PID-5 facets also found facets that have cross-loadings on more than one domain and the number, magnitude and domain location of the loadings for crossloading facets varies across study samples [7].

In an attempt to address the variable nature of PID-5 factor analytic results, Watters and Bagby [7] conducted a meta-analysis of the PID-5 lower-order factor structure. They found that the level of crossloadings decreased when multiple samples were combined, and that the 15 facets scoring version of domains showed the lowest number of cross-loading facets (only anhedonia loaded on two domains substantially). A comparison of the factor structures of the three models is in Table 1.

One of our objective is to compare the factor structure of the 15 and 25 facets scoring version by confirmatory factor analysis. We also intend to quantify whether substantive differences in results occur because of using different domain scoring methods. In order to compare

**Table 1. Fit indexes of CFA of factor structures of PID-5.**

| | Number of items | $\chi^2$ | df | $\chi^2$/df | CFI | RMSEA | SRMR | Cr α |
|---|---|---|---|---|---|---|---|---|
| **Entire Model (5 Domains)** | | | | | | | | |
| APA Model | 123 | 16967.47 | 7355 | 2.31 | .74 | .05 | .07 | |
| Early Krueger Model | 220 | 51262.40 | 23835 | 2.15 | .63 | .04 | .09 | |
| Watters & Bagby Model | 220 | 51149.67 | 23835 | 2.15 | .63 | .04 | .09 | |
| **Domain level** | | | | | | | | |
| **Negative Affectivity** | | | | | | | | |
| APA Model | 23 | 1150.05 | 227 | 5.07 | .84 | .08 | .08 | .91 |
| Early Krueger Model | 53 | 3999.50 | 1318 | 3.03 | .79 | .06 | .08 | .94 |
| Watters & Bagby Model | 46 | 3021.89 | 983 | 3.07 | .83 | .06 | .07 | .94 |
| **Detachment** | | | | | | | | |
| APA Model | 24 | 910.49 | 249 | 3.66 | .87 | .07 | .05 | .89 |
| Early Krueger Model | 46 | 3129.30 | 940 | 3.33 | .81 | .06 | .06 | .92 |
| Watters & Bagby Model | 52 | 4142.02 | 1268 | 3.27 | .78 | .06 | .07 | .93 |
| **Antagonism** | | | | | | | | |
| APA Model | 21 | 755.20 | 186 | 4.06 | .88 | .07 | .06 | .90 |
| Early Krueger Model | 43 | 3048.84 | 855 | 3.56 | .80 | .07 | .08 | .94 |
| Watters & Bagby Model | 53 | 4504.75 | 1319 | 3.41 | .77 | .06 | .08 | .94 |
| **Disinhibition** | | | | | | | | |
| APA Model | 22 | 564.86 | 206 | 2.74 | .92 | .05 | .05 | .90 |
| Early Krueger Model | 46 | 2914.74 | 984 | 2.96 | .81 | .06 | .09 | .88 |
| Watters & Bagby Model | 36 | 2008.40 | 590 | 3.40 | .82 | .06 | .08 | .89 |
| **Psychoticism** | 33 | 2004.94 | 492 | 4.08 | .85 | .07 | .06 | .95 |

Note. N = 588

APA Model: Fifteen facets (three per domain) used in the PID-5 scoring algorithm of Krueger et al. (2013; APA copyright); Early Krueger Model: Facet-domain placement is based on Krueger et al. (2012); Watters & Bagby Model: Facet-domain placement is based on Watters & Bagby (2018); Cr α = Cronbach α, CFI = comparative fit index, RMSEA = Root Mean Square Error Of Approximation; SRMR = Standardized Root Mean Square Residual,; Weak fit indexes are in gray cells.

the different methods used in research examining the psychometric properties of PID-5 with exploratory or confirmatory factor analysis, Table 2 presents an overview of methods for such studies. This mini-review is also a means to compare the results of studies that analyze the psychometric properties of PID-5 in different languages.

We found one study that examined the test-retest reliability of the PID–5. Wright et al. [8] found that across an average of 1.44 years, Cohen's d ranged between .17 (Antagonism) and 0.00 (Negative Affectivity) for the domains whose range is in the little to no change area according to Cohen's guidelines [9] and between little and small change was found for the facets (d = .02 (Depressivity) and -.30 (Submissiveness).

The current study aimed to examine the psychometric properties of the Hungarian PID-5 facet and domain scales among Hungarian-speaking adults from a non-clinical community sample as well as in a Hungarian clinical sample. Specifically, we analyzed its facets and domains' internal consistency and test-retest reliability. In addition, we tested the assumption of whether the DSM-5 traits are indeed maladaptive personality traits by analyzing their correlations with general personality disorder severity measured by the sum of the SCID-II screening questionnaire and general distress measured by SCL-90-R GSI. We also analyzed the specific associations between PID-5 domains and facets with personality disorder dimensions measured by the SCID-II screening questionnaire and the nine symptom dimensions of SCL-

**Table 2. Overview of methods for studies examining the psychometric properties of PID-5 with exploratory or confirmatory factor analysis.**

| Publication | Population | Technique | 15 factors or 25 factors | Item or domain level CFA | Results | Estimation | $\chi^2$ corr | Fit indices |
|---|---|---|---|---|---|---|---|---|
| Bach et al. (2018) Personality Inventory for DSM-5 (PID-5) in Clinical Versus Nonclinical Individuals: Generalizability of Psychometric Features [21] | Danish clinical (n = 598) and non-clinical (n = 598) sample | Exploratory structural equation modeling analyses | 25 factors | - | The results demonstrated acceptable psychometric properties for both samples and supported strong measurement invariance across the groups at the domain level | ML | yes | RMSEA, CFI, SRMSR |
| Bastiaens et al. (2016) The Construct Validity of the Dutch Personality Inventory for DSM-5 Personality Disorders (PID-5) in a Clinical Sample [22] | Flemish inpatients (n = 240) | Exploratory Structural Equation Modeling | 25 factors | - | Our results confirmed the original five-factor structure of the PID-5. The reliability and the convergent and discriminant validity of the PID-5 proved to be adequate | ML | yes | CFI, SRMSR |
| Bo et al. (2016) Reliability and hierarchical structure of DSM-5 pathological traits in a Danish mixed sample [29] | Danish clinical (n = 195) and non-clinical (n = 924) sample | EFA | 15 factors | - | In terms of internal consistency and item discrimination, the applied PID-5 scales were generally found reliable and functional; our data resembled the five-factor structure of previous findings, and we identified a hierarchical structure from one to five factors that was conceptually reasonable and corresponded with existing findings. These results support the new DSM-5 trait model and suggest that it can be generalized to other languages and cultures | ML | ? | ? |
| Coelho et al. (2020) The Arabic Version of the Personality Inventory for the DSM-5 (PID-5) in a Clinical Sample of United Arab Emirates (UAE) Nationals [30] | United Arab Emirates, clinical (n = 156) and non-clinical (n = 156) sample | EFA | 25 factors | - | As expected, the clinical sample presented statistically significantly higher scores than the non-clinical sample, with medium to high effect sizes. In addition, all the PID-5 domains showed positive correlations with most of the symptomatic constellations of the SCL-90-R as well as the PID-5 facets with all their SCL-90-R counterparts. However, our findings did not entirely replicate the PID-5 original 5-factor structure, as only a 4-factor solution was retained. Conclusions: Fu—ture studies with the Arabic PID-5 in clinical samples are needed to understand its relevance and clinical utility in Arabic countries. | ? | ? | ? |

*(Continued)*

**Table 2.** (*Continued*)

| Publication | Population | Technique | 15 factors or 25 factors | Item or domain level CFA | Results | Estimation | $\chi^2$ corr | Fit indices |
|---|---|---|---|---|---|---|---|---|
| De Clercq et al. (2014) The Hierarchical Structure and Construct Validity of the PID-5 Trait Measure in Adolescence [31] | Flemish, healthy adolescents (n = 434) | EFA | 25 factors | - | Results indicate an acceptable reliability for the majority of the PID-5 facets and a tendency toward structural convergence of the adolescent PID-5 structure with the adult proposal. Convergent validity with age-specific facets of personality pathology was generally supported, but discriminant validity appeared to be low. Beyond the findings that support the applicability of the PID-5 in adolescents, developmental issues may be responsible for specific differences in the adolescent PID-5 structure, the rather poor discriminant validity of the PID-5, and the lower reliability of a small number of PID-5 facets. | ML | yes | RMSEA, CFI, SRMSR, BIC |
| De Fruyt et al. (2013) General and Maladaptive Traits in a Five-Factor Framework for DSM-5 in a University Student Sample [32] | Flemish non-clinical sample (n = 240) | EFA | 25 factors | - | A joint factor analysis of, respectively, the NEO domains and their facets with the PID-5 traits showed that general and maladaptive traits are subsumed under an umbrella of five to six major dimensions that can be interpreted from the perspective of the five factor model or the Personality Psychopathology Five. | ML | yes | TLI, RMSEA, SRMR |
| Fang et al. (2021) Personality Inventory for DSM-5 in China: Evaluation of DSM-5 and ICD-11 Trait Structure and Continuity With Personality Disorder Types [33] | Chinese, clinical (n = 406) and non-clinical (n = 3550) sample | Parallel analysis, CFA, correlation and regression analysis | 15 factors | both | Serial CFAs confirmed the rationality of the PID-5's lower-order 25-facet structure and higher-order five-domain structure in both samples. Correlation and regression analyses showed that DSM-5 specified traits explain the variance in PD presentation with a manifold stronger correlation (R 2 = 0.24–0.44) than non-specified traits (R 2 = 0.04–0.12). Overall, the PID-5 was shown to be a reliable, stable, and structurally valid assessment tool that captures pathological personality traits related to DSM-5 and ICD-11 PDs. | MLR | yes | CFI, SRMR, RMSEA |

(*Continued*)

**Table 2.** (*Continued*)

| Publication | Population | Technique | 15 factors or 25 factors | Item or domain level CFA | Results | Estimation | $\chi^2$ corr | Fit indices |
|---|---|---|---|---|---|---|---|---|
| Ferrer et al. (2018) The Psychometric Properties of the Personality Inventory for the DSM-5 (PID-5) in a Colombian Clinic Sample [34] | Colombia, clinical sample (n = 341) | CFA | both | both | Results supported the existence of the 25 first-order factors. In terms of domains (second-order analysis), several organization models were posed. The results supported the model proposed by Krueger, Derringer, Markon, Watson, and Skodol (2012). Men scored significantly higher than women on grandiosity, irresponsibility, manipulativeness, risk-taking, antagonism, and disinhibition. Women scored significantly higher than men on emotional lability and intimacy avoidance. The concurrent validity of PID with the PBQ-SF was high, giving support to the traits of personality disorder models of the DSM-5. | WLS | yes | CFI, NNFI, RMSEA |
| Fossati et al. (2013) Reliability and Validity of the Personality Inventory for DSM-5 (PID-5): Predicting DSM-IV Personality Disorders and Psychopathy in Community-Dwelling Italian Adults [35] | Italian non-clinical sample (n = 710) | Parallel Analysis and CFA | 25 factors | domain level | Parallel analysis and confirmatory factor analysis supported the theoretical five-factor model of the PID-5 trait scales. Regression analyses showed that both PID-5 trait and domain scales explained a substantial amount of variance in the PDQ-4+ PD scales, with the exception of the Passive-Aggressive PD scale. When the PID-5 was administered to a second independent sample of 389 Italian adult community dwelling volunteers, the basic psychometric properties of the scale were replicated. In this second sample, the PID-5 trait and domain scales proved to be significant predictors of psychopathy measures. As a whole, the results of the present study support the hypothesis that the PID-5 is a reliable instrument which is able to recover DSM-IV PDs, as well as to capture personality pathology that is not included in the DSM-IV (namely, psychopathy). | WLS | yes | RMSEA, TLI, CFI, IFI, SRMSR |

(*Continued*)

**Table 2.** (*Continued*)

| Publication | Population | Technique | 15 factors or 25 factors | Item or domain level CFA | Results | Estimation | χ² corr | Fit indices |
|---|---|---|---|---|---|---|---|---|
| Gutierrez et al. (2017) Psychometric Properties of the Spanish PID-5 in a Clinical and a Community Sample [36] | Spanish, clinical (n = 446) and non-clinical (n = 1036) sample | EFA | 25 factors | | Facet scales showed good internal consistency in both samples (median α .86 and .79) and were unidimensional under exploratory and confirmatory approaches. They were also able to distinguish between clinical and community subjects with a mean standardized difference of z = .81. All facets except for Risk Taking were unipolar, such that the upper poles indicated pathology and the lower poles reflected normality, rather than the opposite pole of abnormality. The entire PID-5 hierarchical structure, from one to five factors, was confirmed in both samples with Tucker's congruence coefficients over .95. | ML, Goldberg's bass-ackwards approach | yes | TLI, RMSEA, PGFI |
| Gutierrez et al. (2019) Toward an Integrated Model of Pathological Personality Traits: Common Hierarchical Structure of the PID-5 and the DAPP-BQ [37] | Spanish, psychiatric outpatients (n = 414) | EFA | 25 factors | - | A common hierarchical structure underlies both PID-5 and the DAPP-BQ. Two thirds of the PID-5 and DAPP-BQ facets measure essentially the same traits, although the pairings were not exactly as predicted. Among higher order domains, only PID Negative Affectivity and Detachment converged unambiguously with DAPP Emotional Dysregulation and Inhibition. Overall, the PID-5 and the DAPP-BQ reflect, with small divergences, one and the same structure of pathological personality traits. | Goldberg's Bass-Ackwards method | yes | GFI, AGFI, RMSR, CFI, NNFI, LS |
| Quilty et al. (2013) The Psychometric Properties of the Personality Inventory for DSM-5 in an APA DSM-5 Field Trial Sample [38] | Canada, outpatients (n = 201) | Parallel analysis, Velicer's minimum average partial (MAP) tests | 25 factors | - | The internal consistencies of the PID-5 domain and facet trait scales were acceptable. Results supported the unidimensional structure of all trait scales but one, and the convergence between the PID-5 and analogous NEO PI-R scales. Evidence for discriminant validity was mixed. Overall, the current investigation provides support for the psychometric properties of this diagnostic instrument in psychiatric samples. | RML | | |

(*Continued*)

**Table 2.** (Continued)

| Publication | Population | Technique | 15 factors or 25 factors | Item or domain level CFA | Results | Estimation | $\chi^2$ corr | Fit indices |
|---|---|---|---|---|---|---|---|---|
| Roskam et al. (2015) The Psychometric Properties of the French Version of the Personality Inventory for DSM-5. [39] | French, healthy sample (n = 2532) | Parallel analysis, reliability analysis, EFA | 25 factors | - | The results support the assumption of unidimensionality of both the facets and the domains. Exploratory factor and hierarchical analyses replicated the five-factor structure as initially proposed in the PID-5 | Goldberg's Bass-Ackwards method | yes | CFI, GFI, RMR |
| Shoajei et al. (2020) Psychometric Properties of the Persian Version of Personality Inventory for DSM-5 (PID-5) in Psychiatric Patients [40] | Persian, psychiatric patients (n = 400) | CFA | 25 factors | both | Adequate internal consistency coefficients were obtained for domains and facets. In addition, the test-retest coefficients (up to 0.70) suggested scale stability. Confirmatory factor analysis supported the original five-factor model of the inventory. The convergent validity of the inventory with the TCI-R scale was appropriate. The results of the study supported the psychometric properties of the Persian version of PID-5 in psychiatric populations. | ? | yes | CFI, TLI, RMSEA |
| Somma et al. (2017) Reliability, factor structure, and associations with measures of problem relationship and behavior of the personality inventory for DSM-5 in a sample of italian community-dwelling adolescents [41] | Italian, healthy adolescents (n = 1264) | Exploratory structural equation modeling analyses, weighted least square mean and variance adjusted (WLSMV) confirmatory factor analyses | 15 factors | - | Exploratory structural equation modeling analyses provided moderate support for the a priori model of PID-5 trait scales. Ordinal logistic regression analyses showed that selected PID-5 trait scales predicted a significant, albeit moderate (Cox & Snell R2 values ranged from .08 to .15, all ps < .001) amount of variance in Questionnaire on Relationships and Substance Use variables. | MLR, AIC, BIC, SABIC | yes | RMSEA, TLI, CFI |
| Thimm et al. (2017) Hierarchical Structure and Cross-Cultural Measurement Invariance of the Norwegian Version of the Personality Inventory for DSM–5 [42] | Norwegian non-clinical sample (n = 503) | EFA | 15 factors | - | The 5-factor structure was generally congruent with international findings, and support for measurement invariance across the Norwegian and a matched U.S. sample was found. Conclusively, the results indicate that scores on the Norwegian PID–5 have sound psychometric properties, which are substantially comparable with the original U.S. version, supporting its use in a Norwegian population. | MLR | yes | RMSEA, CFI |
| Thomas et al. (2013) The Convergent Structure of DSM-5 Personality Trait Facets and Five-Factor Model Trait Domains [43] | USA, non-clinical sample (n = 808) | EFA, parallel analysis | 25 factors | - | Results indicate that the five higher-order factors of the conjoint EFA reflect the domains of the Five-Factor Model of normative personality (FFM). The authors briefly discuss implications of this correspondence between the normative FFM and the pathological PID-5. | ML | ? | ? |

*(Continued)*

**Table 2.** (Continued)

| Publication | Population | Technique | 15 factors or 25 factors | Item or domain level CFA | Results | Estimation | $\chi^2$ corr | Fit indices |
|---|---|---|---|---|---|---|---|---|
| Van den Broeck et al. (2014) Hierarchical Structure of Maladaptive Personality Traits in Older Adults: Joint Factor Analysis of the PID-5 and the DAPP-BQ [26] | Flemish, healthy adults (n = 173) | PCA | 25 factors | - | A joint hierarchical factor analysis showed clear convergence between four PID-5 dimensions (Negative Affect, Detachment, Antagonism, Disinhibition) and conceptually similar DAPP-BQ components. Moreover, the PID-5 and the DAPP-BQ showed meaningful associations on different levels of their joint hierarchical factor structure. | - | - | - |
| Wright et al. (2012) The Hierarchical Structure of DSM-5 Pathological Personality Traits [44] | USA, non-clinical sample (n = 2461) | EFA | 25 factors | - | Exploratory factor analysis replicated the initially reported five-factor structure as indicated by high factor congruencies. The two-, three-, and four- factor solutions estimated in the hierarchy of the DSM-5 traits bear close resemblance to existing models of common mental disorders, temperament, and personality pathology. Thus, beyond the description of individual differences in personality disorder, the trait dimensions might provide a framework for the metastructure of psychopathology in the DSM-5 and the integration of a number of ostensibly competing models of personality trait covariation. | ML | ? | ? |
| Zimmermann et al. (2014) The structure and correlates of self-reported DSM-5 maladaptive personality traits: findings from two German-speaking samples [18] | German, clinical (n = 212) and non-clinical (n = 577) sample | 25 item-level CFAs based on the polychoric correlation matrix and robust weighted least square estimation, EFA | 25 factors | Item-level CFA, facet-level EFA | (a) the factor structure of DSM-5 trait facets is largely in line with the proposed trait domains of Negative Affectivity, Detachment, Antagonism, Disinhibition, and Psychoticism; (b) all DSM5 trait domains except Psychoticism are highly related to the respective domains of the Five-Factor Model of personality; (c) the trait facets are positively associated with a self-report measure of general personality dysfunction; and (d) the DSM-5 trait facets show differential associations with a range of self-reported DSM-IV Axis I disorders. These findings give further support to the new DSM-5 trait model and suggest that it may generalize to other languages and cultures. | ML | no | CFI, TLI, RMSEA |

Notes: AGFI: adjusted goodness of fit index, AIC: Akaike Information Criterion, BIC: Bayesian information criterion, CFA: comparatory factor analysis, (R)CFI: (robust) comparative fit index, EFA: exploratory factor analysis, GFI: goodness of fit index, IFI: incremental fit index, LS: loading simplicity index, ML: maximum likelihood estimation, MLR: robust maximum likelihood estimation, NNFI: non-normed fit index, PCA: principal component analysis, RMSEA: root mean squared error of approximation, RMSR: Root mean squared residual, SABIC: sample size adjusted BIC, SRMSR: Standardized root mean squared residual, TLI: Tucker-Lewis Index, WLS: weighted least square estimation, WLSMV: weighted least square mean and variance adjusted method

90-R. Finally, we analyzed the associations between PID-5 traits and age, gender, and education.

## 2. Methods

### 2.1. Participants and procedure

471 non-clinical participants were recruited by 20 psychology students who received class credit for their work. Each student aimed to collect data by using snowball sampling techniques by reaching around 50 volunteers with no known psychiatric disorders. Data was also gathered from 314 clinical patients of Semmelweis University's Department of Psychiatry and Psychotherapy. Participants were stratified according to gender and age range. The majority of the non-clinical participants were female (69,6%) and their mean age was 37.63 years (SD: 12.96, their ages were between 18 and 74 years). The majority of the clinical participants were female as well (69,7%) and their mean age was 37,41 years (SD: 13,34, their ages were between 18 and 74 years). Every participant was Caucasian.

Based on SCID II interview, 173 participants (55,1%) in the clinical sample had at least one diagnosis of personality disorder, 117 subjects (37,3%) had multiple diagnoses (two or more personality disorders). No information was available on the clinical history of 64 individuals. The distribution of the different types of personality disorder was as follows:

Avoidant 21,6% (54 subjects); Dependent 16,06% (40 subjects); Obsessive-compulsive 18,145% (45 subjects); Passive-aggressive 13,2% (33 subjects); Depressive 24,4% (61 subjects); Paranoid 14,8% (37 subjects); Schizotype 2,4% (6 subjects); Schizoid 1,2% (3 subjects); Histrionic 14% (35 subjects); Narcissistic 17,6% (44 subjects) and Borderline 30,8% (77 subjects).

Participants completed all questionnaires online in Google-docs, a secure online survey tool. Each participant gave written consent to participate by choosing the "agree with" option in response to an informed consent form.

In the clinical sample some of the data was missing, and these missing data were not replaced, therefore the sample sizes vary in each analysis. There were 92 individuals who did not mark an answer for at least one item of the PID-5 questionnaire. One participant was excluded from the analysis due to too much missing data (50%), for the others, the number of missing data did not exceed 10%. The sample size was the lowest (222) in the analyzes where all PID-5 items were included in the confirmatory factor analysis (CFA). In CFAs when the complete factor structure was analyzed, the sample sizes varied between 222–241. Sample sizes ranged from 295 to 307 for item-level analyzes and from 265 to 277 for domain-level analyzes. No data were missing in the analyzes on facet- and domain scales, as the scores on the scales were calculated by averaging.

In addition, 93 (62 female) non-clinical subjects participated in the test-retest reliability study. The age of the participants in the test-retest reliability study varied from 19 to 62 years (Mean: 40.32; SD: 13.08). The descriptive statistics of the samples can be found in Table 3. The local Ethical Committee of the ELTE PPK (2018/8) approved the study protocol. The Regional and Institutional Committee of Science and Research Ethics of Semmelweis University approved the research procedure.

### 2.2. Measures

**2.2.1. Personality Inventory for the DSM-5 (PID-5).** The PID-5 is a 220-item inventory that assesses personality disorder traits according to the DSM-5 trait model [2]. Items are rated on a 4-point Likert-type scale from 0 (Very False or Often False) to 3 (Very True or Often True). The PID-5 yields scale scores for 25 facets and the following five domains: Negative Affectivity, Detachment, Antagonism, Disinhibition, and Psychoticism. The official

**Table 3. Descriptive statistics of the samples.**

| | N | Age (year) | | | Educational level | | | | |
|---|---|---|---|---|---|---|---|---|---|
| | | M (SD) | Min | Max | Elementary | Vocational | High school | College | University |
| **Non clinical sample** | | | | | | | | | |
| **Total** | 588 | 37.49 (13.22) | 18 | 74 | 3 (.5%) | 13 (2.2%) | 252 (42.9%) | 165 (28.1%) | 155 (26.4%) |
| **Male** | 251 | 37.18 (13.66) | 19 | 74 | 2 (.8%) | 10 (4.0%) | 104 (41.4%) | 58 (23.1%) | 77 (30.7%) |
| **Female** | 337 | 37.73 (12.90) | 18 | 74 | 1 .3%) | 3 (.9%) | 148 (43.9%) | 107 (31.8%) | 78 (23.1%) |
| **Non clinical retest sample** | | | | | | | | | |
| **Total** | 93 | 40.32 (13.08) | 19 | 62 | 0 (0%) | 1 (1.1%) | 32 (34.4%) | 31 (33.3%) | 29 (31.2%) |
| **Male** | 31 | 40.13(13.82) | 20 | 62 | 0 (0%) | 1 (3.2%) | 7 (22.6%) | 12 (38.7%) | 11 (35.5%) |
| **Female** | 62 | 40.42(12.8) | 19 | 60 | 0 (0%) | 0 (0%) | 25 (40.3%) | 19 (30.6%) | 18 (29%) |
| **Clinical sample** | | | | | | | | | |
| **Total** | 314 | 36.99 (12.77) | 18 | 74 | 2 (.6%) | 61 (19.4%) | 146 (46.5%) | 46 (14.6%) | 53 (16.9%) |
| **Male** | 95 | 36.03 (11.34) | 18 | 63 | 0 (0%) | 19 (20.0%) | 49 (51.6%) | 14 (14.7%) | 12 (12.6%) |
| **Female** | 219 | 37.41 (13.34) | 18 | 74 | 2 (.9%) | 42 (19.2%) | 97 (44.3%) | 32 (14.6%) | 41 (18.7%) |

Note. M = mean; SD = Standard Deviation

Hungarian translation used in the current study has been permitted by APA. Two independent translators translated the PID-5 into Hungarian, then a group of experts compared the two translations and agreed on a first version. We distributed the first version among patients of the Department of Psychotherapy at Semmelweis University, who commented on it, and based on their comments, we made corrections to the translation. There was no back translation made. A Hungarian version is freely available at the following site: https://lelekbenotthon.hu/pid-5-dsm-5-szemelyisegkerdoiv/

**2.2.2. The Structured Clinical Interview for DSM-IV Axis II Personality Questionnaire (SCID-II PQ).** SCID-II PQ [10, 11] contains 119 items in a yes/no response format, corresponding to the DSM-IV personality disorders' criteria. Regional and Institutional Committee of Science and Research Ethics of Semmelweis University recommended to leave out items referring to antisocial personality disorder, therefore we used only 104 items. In the present study, the PDs were expressed dimensionally by averaging the scores (range 0–1) of each SCID-II-PQ item per PD. Higher scores are reflecting more of the personality characteristic being assessed. We followed Hopwood et al.'s [12] method and eschewed categorical PDs in favor of continuous symptom counts for this study. In our non-clinical sample, where participants do not fit to a full PDs diagnosis, the use of continuous scales of PDs seemed to be a viable method. The SCID-II screening questionnaire has good reliability and validity data [13]. For the current study, we calculated PD scores for those PD-s which are proposed to be retained in DSM-5—except for Antisocial Personality Disorder -, as well as a total score of SCID-II by averaging. We did not measure the symptoms of Antisocial Personality Disorder since our ethical committee did not allow us to use the antisocial modul of the SCID-II-PQ for legal and ethical reasons. The internal consistency of SCID-II-PQ scales ranged from $\alpha = .55$ (schizoid) to $\alpha = .89$ (borderline).

**2.2.3. Symptom Check List-90-Revised (SCL-90-R).** In order to assess the level of general distress and nine specific symptom dimensions (Somatization (SOM); Obsessive-compulsive (O-C); Interpersonal sensitivity (I-S); Depression (DEP); Anxiety (ANX); Hostility (HOS); Phobic anxiety (PHOB); Paranoid ideation (PAR); and Psychoticism (PSY), the Hungarian version of the SCL-90-R [14] was used. For analysis, a Global Severity Index (GSI) and the subscale scores were calculated by averaging. The internal consistency of these scales ranged from $\alpha = .82$ (anger-hostility) to $\alpha = .95$ (depression).

## 2.3. Data analytic procedure

MANOVA was run to analyze gender differences along with the 25 personality disorder traits, Pearson and Spearman's correlation were calculated to evaluate the relationship between PID5 traits and age as well as educational level.

The reliability of the Hungarian PID-5 was examined by calculating the internal consistencies of the facet and domain scores (Cronbach's alpha). We examined the 1-month test-retest reliabilities for the PID–5 scales by using Pearson correlation coefficients and Cohen's d, which can establish a magnitude of change within this time interval. According to Cohen [9] d of .2 - .5 means small change, d of .5 - .8 means medium change, and if the value of d is higher than 0.8 the change can be considered large.

Confirmatory factor analysis (CFA) was conducted to test the unidimensionality of the PID-5 scales and to explore the factor structure of PID-5 facets and domains. Model fit was evaluated using the usual fit indexes: Comparative Fit Index (CFI), Root Mean Square Error of Approximation (RMSEA), Standardized Root Mean Square Residual (SRMR). The CFI is acceptable if the value is higher than .90, the RMSEA and SRMR indexes should be less than .10 for adequate fit [15–17].

To examine the validity of the Hungarian PID-5, relationships of PID-5 scales with scales of SCID II PQ and SCL90-R were explored using correlation analysis.

## 3. Results

### 3.1. Reliability

**3.1.1. Internal consistency.**   Internal consistency results are reported in Tables 4 and 5. Alpha coefficients for the Hungarian PID-5 domain scores ranged from .88 (Detachment) to .95 (Psychoticism) in the non-clinical sample, and from .90 (Antagonism) and .95 (Psychoticism) in the clinical sample. For the facets, scores ranged from .53 (Suspiciousness) to .95 (Eccentricity) in the non-clinical sample, and from .71 (Suspiciousness and Restricted Affectivity) to .95 (Eccentricity) in the clinical sample. The mean alpha for the domain scores was .90 (non-clinical) and .92 (clinical) and .81 (non-clinical) and .84 (clinical) for the facet scores. In summary, most scales have appropriate reliability, except for the Suspiciousness facet in the non-clinical sample, the reliability of which is low.

**3.1.2. Test-retest reliability and magnitudes of change.**   As it is shown in Table 4, internal reliability coefficients of PID-5 dimensions on test-retest sample were between .64 for Irresponsibility, .96 for Eccentricity. Test-retest correlation coefficients (Pearson) were between .53 (Callousness) and .86 (Risk-taking), which reflects a moderate and strong connection between the two measurements. Utilizing Cohen's d to establish magnitude of change within a one month interval, a small amount of change (i.e., d of .2–.4; Cohen, 1988) was observed in Negative Affectivity (d = −.30), and in Disinhibition (d = −.20) domains, and in Anxiousness (d = −.22), Separation Insecurity (d = −.21), Withdrawal (d = −.22), Restricted Affectivity (d = −.23), Distractibility (d = −.33). In other domains and facets, we found a smaller d than−.20, which reflects little to no change according to Cohen's guidelines (Cohen, 1988).

**3.1.3. Confirmatory factor analysis.**   We ran a Confirmatory Factor Analysis in order to explore the factor structure of PID-5 traits; results are reported in Tables 4 and 5. We analyzed the fitting of all the three factor structures found in the literature (the APA model [1], the early Krueger model [2] and the Watters-Bagby model [7]). The results showed that the RMSE and SRMR fit indices reach the good or acceptable level in both samples in case of all the three models, while CFI scores indicate a poor fit in all samples and models in almost all domains. Based on the APA model, the Disinhibition scale fits well in both samples, and Detachment

**Table 4. Internal consistency and item level CFA of Personality Inventory for DSM-5 facets and domains.**

| | Number of items | Cr α | MII | MIT | CFI | RMSEA | SRMR | Test-Retest | | |
| --- | --- | --- | --- | --- | --- | --- | --- | --- | --- | --- |
| | | | | | | | | Cr α | corr. | Cohen d |
| **Negative Affectivity** | 23 | .91 | .30 | .30 | .84 | .08 | .08 | .89 | .81** | .30** |
| **Detachment** | 24 | .89 | .27 | .27 | .87 | .07 | .05 | .89 | .79** | .07 |
| **Antagonism** | 21 | .90 | .31 | .31 | .88 | .07 | .06 | .91 | .71** | .11 |
| **Disinhibition** | 22 | .90 | .28 | .28 | .92 | .05 | .05 | .87 | .71** | .20* |
| **Psychoticism** | 33 | .95 | .35 | .35 | .85 | .07 | .06 | .95 | .68** | .20 |
| Anhedonia | 8 | .80 | .37 | .54 | .94 | .08 | .05 | .74 | .77 | .03 |
| Anxiousness | 9 | .89 | .48 | .65 | .95 | .09 | .03 | .88 | .82** | .22* |
| Attention Seeking | 8 | .87 | .45 | .63 | .92 | .11 | .05 | .88 | .73** | .11 |
| Callousness | 14 | .84 | .31 | .51 | .87 | .09 | .05 | .81 | .53** | .09 |
| Deceitfulness | 10 | .83 | .65 | .54 | .94 | .08 | .04 | .85 | .68** | .08 |
| Depressivity | 14 | .90 | .44 | .62 | .86 | .11 | .06 | .87 | .72** | .12 |
| Distractibility | 9 | .86 | .41 | .59 | .95 | .07 | .03 | .79 | .75** | .33** |
| Eccentricity | 13 | .95 | .59 | .75 | .96 | .10 | .04 | .96 | .74** | .20 |
| Emotional Lability | 7 | .77 | .32 | .49 | .68 | .22 | .13 | .80 | .78** | .19 |
| Grandiosity | 6 | .80 | .40 | .56 | .98 | .06 | .03 | .83 | .70** | .09 |
| Hostility | 10 | .83 | .33 | .52 | .89 | .09 | .05 | .81 | .78** | .14 |
| Impulsivity | 6 | .82 | .44 | .59 | .98 | .06 | .02 | .85 | .70** | .11 |
| Intimacy Avoidance | 6 | .67 | .28 | .42 | .97 | .06 | .03 | .71 | .73** | .06 |
| Irresponsibility | 7 | .68 | .23 | .39 | .96 | .05 | .03 | .64 | .68** | .06 |
| Manipulativeness | 5 | .75 | .39 | .53 | .94 | .12 | .04 | .75 | .72** | .13 |
| Separation Insecurity | 7 | .81 | .38 | .55 | .95 | .09 | .04 | .79 | .66** | .21* |
| Submissiveness | 4 | .78 | .47 | .58 | .99 | .03 | .01 | .80 | .63** | .03 |
| Perceptual Dysregulation | 12 | .84 | .34 | .53 | .90 | .08 | .05 | .81 | .63** | .19 |
| Perseveration | 9 | .82 | .34 | .53 | .89 | .10 | .05 | .77 | .71** | .17 |
| Restricted Affectivity | 7 | .79 | .35 | .52 | .91 | .10 | .05 | .80 | .67** | .23* |
| Rigid Perfectionism | 10 | .87 | .41 | .60 | .93 | .08 | .04 | .88 | .70** | .14 |
| Risk Taking | 14 | .88 | .35 | .55 | .82 | .11 | .08 | .90 | .86** | .16 |
| Suspiciousness | 7 | .54 | .17 | .30 | .78 | .07 | .05 | .66 | .68** | .10 |
| Unusual Beliefs | 8 | .78 | .33 | .50 | .83 | .14 | .07 | .81 | .62** | .11 |
| Withdrawal | 10 | .87 | .41 | .60 | .88 | .12 | .06 | .89 | .79** | .22* |

Note. N = 588; $N_{retest}$ = 93

Cr α = Cronbach α, MII = mean inter-item correlations, MIT = mean item-total correlations; CFI = comparative fit index, RMSEA = Root Mean Square Error Of Approximation; SRMR = Standardized Root Mean Square Residual, corr. = Pearson correlation coefficients; Weak fit indexes are in gray cells.

* p < .05

** p < .01.

fits well in the clinical sample. According to the results of item-level CFA, CFI of the PID-5 facet scales ranged from .81 to .99 in the non-clinical sample, and from .63 to .99 in the clinical sample. The values of RMSEA ranged from .01 to .22, and the SRMR ranged from .01 to .13 in the non-clinical sample, and in the clinical sample RMSEA ranged from .04 to .15, while the SRMR ranged from .01 to .15. (For more detail, see Tables 4 and 5.) These results indicate poor fit of 11 out of the 25 PID-5 facets in the non-clinical sample, and of 13 out of the 25 facets in the clinical sample. The scales that are found to fit the least in both samples are Emotional Lability, Unusual Beliefs, Depressivity, Perseveration, Restricted Affectivity and Risk Taking.

**Table 5. Internal consistency and item level CFA of Personality Inventory for DSM-5 scales on clinical samples.**

| | | Cr α | CFI | RMSEA | SRMR |
|---|---|---|---|---|---|
| | N of items | CL OV | CL OV | CL OV | CL OV |
| **Negative Affectivity** | 23 | .91 | .81. | .09 | .08 |
| **Detachment** | 24 | .91 | .90 | .06 | .06 |
| **Antagonism** | 21 | .89 | .80 | .09 | .07 |
| **Disinhibition** | 22 | .91 | .93 | .06 | .06 |
| **Psychoticism** | 33 | .95 | .86 | .07 | .07 |
| Anhedonia | 8 | .87 | .97 | .07 | .04 |
| Anxiousness | 9 | .88 | .90 | .11 | .05 |
| Attention Seeking | 8 | .88 | .94 | .11 | .05 |
| Callousness | 14 | .86 | .86 | .09 | .06 |
| Deceitfulness | 10 | .84 | .87 | .11 | .06 |
| Depressivity | 14 | .91 | .87 | .10 | .06 |
| Distractibility | 9 | .87 | .95 | .09 | .05 |
| Eccentricity | 13 | .95 | .95 | .09 | .04 |
| Emotional Lability | 7 | .80 | .64 | .24 | .14 |
| Grandiosity | 6 | .79 | .99 | .01 | .02 |
| Hostility | 10 | .87 | .95 | .07 | .04 |
| Impulsivity | 6 | .90 | .99 | .04 | .02 |
| Intimacy Avoidance | 6 | .77 | .95 | .11 | .05 |
| Irresponsibility | 7 | .76 | .99 | .01 | .03 |
| Manipulativeness | 5 | .78 | .93 | .14 | .04 |
| Separation Insecurity | 7 | .86 | .94 | .11 | .05 |
| Submissiveness | 4 | .86 | .99 | .04 | .01 |
| Perceptual Dysregulation | 12 | .86 | .88 | .08 | .06 |
| Perseveration | 9/6 | .80 | .75 | .14 | .08 |
| Restricted Affectivity | 7 | .73 | .89 | .09 | .05 |
| Rigid Perfectionism | 10 | .87 | .96 | .06 | .04 |
| Risk Taking | 14 | .86 | .86 | .09 | .07 |
| Suspiciousness | 7 | .71 | .94 | .07 | .04 |
| Unusual Beliefs | 8 | .85 | .86 | .15 | .07 |
| Withdrawal | 10 | .88 | .92 | .09 | .05 |

### 3.2. Validity

**3.2.1. Correlations between PID-5 traits and SCID-II as well as SCL-90-R scores.** Correlations between PID-5 traits and SCID-II personality disorder scores and SCL-90-R scores are reported in Tables 6 and 7. We found that correlations of PID-5 domains and SCID-II total PD score are significant in both the clinical and the non-clinical groups with a Bonferroni adjusted p-value. Effect sizes are moderate or strong. Correlations between the traits and SCID-II total score are stronger in the clinical sample, where correlations are moderate to strong, except for weak correlations with Intimacy Avoidance, Submissiveness and Risk Taking. In the non-clinical sample, correlations between the traits and SCID-II total score are mostly moderate, with the exceptions of avoidance, Submissiveness, Risk Taking and Rigid Perfectionism, Restricted Affectivity and Attention Seeking, which are also weakly correlated.

Different personality disorder scores of SCID-II-PQ are significantly correlated to the expected PID-5 domains and traits with moderate or strong effect size, however, the correlations are stronger in the clinical sample. There are only a few exceptions: Intimacy Avoidance

**Table 6. Pearson correlations between PID-5 traits, SCID-II total personality disorder score and DSM-IV personality disorders proposed to be retained in DSM-5 measured by SCID-II.**

| | | SCID II. Total PD | | STPD | | BPD | | NPD | | AVPD | | OCPD | |
|---|---|---|---|---|---|---|---|---|---|---|---|---|---|
| | | non-clin. | clin. | non-clin. | clin. | non-clin. | clin. | non-clin. | clin. | non-clin. | clin. | non-clin. | clin. |
| Negative Affectivity | APA Model | .45* | .66* | .30* | .42* | .40* | .58* | .26* | .32* | .31* | .39* | .23* | .29* |
| | Early Krueger Model | .49* | .74* | .33* | .47* | .40* | .65* | .32* | .40* | .37* | .40* | .32* | .35* |
| | Watters & Bagby Model | .45* | .68* | .29* | .42* | .34* | .58* | .25* | .33* | .35* | .33* | .32* | .40* |
| Detachment | APA Model | .40* | .48* | .27* | .29* | .31* | .39* | .26* | .23* | .44* | .55* | .20* | .18* |
| | Early Krueger Model | 45* | .62* | .30* | 40* | .36* | .52* | .31* | .31* | .43* | .58* | .23* | .27* |
| | Watters & Bagby Model | .46* | .63* | .30* | .40* | .34* | .51* | .33* | .32* | .43* | .58* | .25* | .26* |
| Antagonism | APA Model | .40* | .46* | .32* | .26* | .31* | .41* | .46* | .59* | .12 | .05 | .24* | .13 |
| | Early Krueger Model | .41* | .51* | .30* | .28* | .32* | .45* | .46* | .64* | .11 | .07 | .24* | .16 |
| | Watters & Bagby Model | .43* | .58* | .31* | .32* | .35* | .52* | .47* | .65* | .12 | .11 | .27* | .21* |
| Disinhibition | APA Model | .44* | .65* | .31* | .38* | .43* | .64* | .31* | .43* | .28* | .35* | .19* | .15 |
| | Early Krueger Model | .47* | .70* | .33* | .42* | .39* | .68* | .34* | .49* | .29* | .32* | .33* | .32* |
| | Watters & Bagby Model | .45* | .62* | .33* | .37* | .42* | .64* | .33* | .44* | .28* | .27* | .22* | .16 |
| Psychoticism | | .47* | .67* | .45* | .63* | .41* | .61* | .44* | .47* | .23* | .32* | .24* | .26* |
| PID-5 traits | Anhedonia | .37* | .45* | .20* | .19* | .33* | .37* | .20* | .16 | .39* | .49* | .13* | .16 |
| | Anxiousness | .45* | .61* | .25* | .36* | .37* | .50* | .25* | .30* | .38* | .44* | .27* | .33* |
| | Attention Seeking | .26* | .43* | .19* | .21* | .22* | .37* | .28* | .54* | .01 | -.05 | .14* | .16 |
| | Callousness | .37* | .43* | .23* | .25* | .30* | .38* | .45* | .48* | .17* | .24* | .20* | .14 |
| | Deceitfulness | .39* | .40* | .28* | .17* | .32* | .39* | .40* | .47* | .20* | .11 | .20* | .06 |
| | Depressivity | .44* | .57* | .26* | .37* | .39* | .54* | .26* | .21* | .37* | .49* | .18* | .22* |
| | Distractibility | .44* | .57* | .31* | .32* | .43* | .51* | .26* | .31* | .36* | .40* | .17* | .15 |
| | Eccentricity | .46* | .66* | .35* | .53* | .42* | .59* | .43* | .48* | .24* | .32* | .24* | .27* |
| | Emotional Lability | .34* | .60* | .29* | .44* | .34* | .61* | .21* | .25* | .17* | .31* | .16* | .26* |
| | Grandiosity | .34* | .42* | .28* | .32* | .25* | .28* | .43* | .57* | .10 | .06 | .23* | .24* |
| | Hostility | .39* | .63* | .25* | .38* | .34* | .59* | .36* | .51* | .11 | .25* | .31* | .33* |
| | Impulsivity | .29* | .56* | .22* | .33* | .29* | .62* | .22* | .38* | .11 | .21* | .15* | .19* |
| | Intimacy Avoidance | .25* | .20* | .19* | .13 | .17* | .17* | .21* | .14 | .15* | .17* | .17* | .06 |
| | Irresponsibility | .37* | .45* | .24* | .29* | .35* | .44* | .31* | .38* | .24* | .26* | .15* | .03 |
| | Manipulativeness | .31* | .35* | .23* | .17* | .25* | .36* | .36* | .44* | .05 | -.03* | .20* | .04 |
| | Perceptual Dysregulation | .41* | .59* | .35* | .57* | .35* | .57* | .35* | .36* | .24* | .34* | .19* | .23* |
| | Perseveration | .44* | .62* | .30* | .37* | .33* | .52* | .26* | .37* | .33* | .40* | .33* | .41* |
| | Restricted Affectivity | .29* | .33* | .19* | .22* | .19* | .27* | .28* | .23* | .28* | .34* | .24* | .15 |
| | Rigid Perfectionism | .26* | .46* | .16* | .27* | .11 | .34* | .20* | .29* | .16* | .24* | .40* | .59* |
| | Risk Taking | .28* | .19* | .27* | .12 | .18* | .27* | .25* | .22* | .15* | -.11 | .25* | .07 |
| | Separation Insecurity | .32* | .43* | .20* | .24* | .28* | .34* | .18* | .23* | .20* | .24* | .14* | .13 |
| | Submissiveness | .17* | .27* | .12 | .18* | .11 | .23* | .03 | .04 | .27* | .30* | .12 | .08 |
| | Suspiciousness | .32* | .64* | .23* | .45* | .19* | .48* | .30* | .44* | .18* | .36* | .22* | .35* |
| | Unusual Beliefs | .36* | .51* | .48* | .61* | .29* | .45* | .35* | .39* | .12 | .18* | .21* | .20* |
| | Withdrawal | .30* | .48* | .23* | .35* | .21* | .38* | .21* | .24* | .42* | .63* | .17* | .21* |

Note. N_nonclinical = 471, N_clinical = 314.

* = p < .005 (Bonferroni corrected level of significance), correlations > .40 are in bold. PD = Personality Disorder, STPD = Schizotypal PD, BPD = Borderline PD, NPD = Narcissistic PD, AVPD = Avoidant PD, OCPD = Obsessive-Compulsive PD. Traits that are proposed as criteria for each retained PD in the DSM-5 are in gray cells.

**Table 7. Pearson correlations between PID-5 traits and SCL90-R scales.**

| | SCL90R | GSI | | SOM | | O-C | | I-S | | DEP | | ANX | | HOS | | PHOB | | PAR | | PSY | |
|---|---|---|---|---|---|---|---|---|---|---|---|---|---|---|---|---|---|---|---|---|---|
| | | non clin | clin | non clin | clin | non clin | clin | non clin | clin | non clin | clin | non clin | clin | non clin | Clin | non clin | clin | non clin | clin | non clin | clin |
| **Negative Affectivity** | APA Model | .65* | .69* | .44* | .40* | .60* | .65* | .58* | .61* | .63* | .68* | .66* | .59* | .50* | .50* | .55* | .58* | .46* | .50* | .54* | .51* |
| | Early Krueger Model | .66* | .71* | .43* | .41* | .65* | .67* | .61* | .66* | .62* | .66* | .63* | .57* | .52* | .56* | .55* | .55* | .52* | .58* | .58* | .61* |
| | Watters & Bagby Model | .64* | .67* | .43* | .38* | .62* | .68* | .58* | .64* | .61* | .65* | .62* | .55* | .49* | .47* | .53* | .52* | .50* | .55* | .54* | .54* |
| **Detachment** | APA Model | .52* | .50* | .35* | .33* | .51* | .36* | .53* | .47* | .48* | .51* | .40* | .41* | .34* | .31* | .45* | .34* | .44* | .37* | .52* | .47* |
| | Early Krueger Model | .62* | .62* | .41* | 40* | .59* | .49* | .61* | .61* | .59* | .62* | .49* | .50* | .40* | .43* | .52* | .46* | .53* | .54* | .60* | .60* |
| | Watters & Bagby Model | .57* | .64* | .36* | .39* | .57* | .48* | .58* | .59* | .53* | .59* | .45* | .48* | .37* | .42* | .48* | .44* | .51* | .52* | .58* | .59* |
| **Antagonism** | APA Model | .35* | .25* | .22* | .09 | .30* | .15 | .28* | .18* | .26* | .18* | .31* | .20* | .36* | .41* | .35* | .16* | .38* | .28* | .38* | .34* |
| | Early Krueger Model | .35* | .27* | .20* | .01 | .32* | .17* | .29* | .22* | .27* | .21* | .31* | .22* | .37* | .43* | .33* | .18* | .37* | .32* | .37* | .37 |
| | Watters & Bagby Model | .39* | .33* | .23* | .14 | .35* | .23* | .32* | .28* | .30* | .25* | .35* | .26* | .43* | .50* | .36* | .23* | .39* | .39* | .40* | .42* |
| **Disinhibition** | APA Model | .52* | .54* | .37* | .32* | .54* | .51* | .45* | .45* | .48* | .47* | .45* | .42* | .48* | .52* | .44* | .40* | .39* | .43* | .36* | .53* |
| | Early Krueger Model | .55* | .56* | .36* | .31* | .56* | .54* | .48* | .48* | .48* | .45* | .48* | .42* | .49* | .54* | .46* | .39* | .44* | .51* | .47* | .56* |
| | Watters & Bagby Model | .52* | .48* | .34* | .29* | .54* | .44* | .45* | .39* | .46* | .39* | .44* | .37* | .47* | .51* | .45* | .35* | .40* | .42* | .46* | .50* |
| **Psychoticism** | | .53* | .61* | .35* | .43* | .52* | .54* | .48* | .52* | .45* | .43* | .43* | .44* | .46* | .57* | .51* | .47* | .47* | .63* | .53* | .65* |
| **PID-5 traits** | Anhedonia | .56* | .55* | .40* | .31* | .53* | .49* | .53* | .49* | .60* | .67* | .45* | .47* | .35* | .30* | .43* | .40* | .42* | .32* | .50* | .45* |
| | Anxiousness | .60* | .63* | .39* | .36* | .58* | 60* | .53* | .58* | .61* | .63* | .65* | .59* | .41* | .40* | .50* | .55* | .41* | .45* | .48* | .48* |
| | Attention Seeking | .24* | .19* | .09 | .03 | .23* | .17* | .19* | .17* | .23* | .17* | .24* | .16 | .26* | .28* | .19* | .11 | .21* | .25* | .24* | .25* |
| Callousness | .28* | .28* | .19* | .15 | .28* | .14 | .26v | .26* | .17* | .16* | .19* | .21* | .32* | .42* | .27* | .20* | .32* | .35* | .33* | .40* | |
| | Deceitfulness | .37* | .18* | .22* | .02 | .32* | .10 | .31* | .16 | .29* | .16 | .33* | .15 | .37* | .36* | .34* | .12 | .37* | .18* | .40* | .26* |
| | Depressivity | .67* | .64* | .44* | .36* | .61* | .54* | .65* | .60* | .69* | .67* | .56* | .48* | .42* | .44* | .57* | .47* | .51* | .69* | .63* | .56* |
| | Distractibility | .53* | .58* | .34* | .38* | .57* | .61* | .46* | .49* | .51* | .56* | .46* | .46* | .42* | .46* | .45* | .43* | .39* | .41* | .47* | .48* |
| | Eccentricity | .47* | .55* | .26* | .32* | .47* | .49* | .46* | .50* | .42* | .42* | .40* | .37* | .39* | .50* | .46* | .39* | .44* | .59* | .46* | .57* |
| | Emotional Lability | .47* | .59* | .32* | .38* | .43* | .54* | .43* | .54* | .47* | .54* | .47* | .46* | .41* | .52* | .39* | .44* | .32* | .48* | .38* | .44* |
| | Grandiosity | .30* | .25* | .19* | .15 | .26* | .16* | .26* | .19* | .21* | .17* | .23* | .19* | .30* | .29* | .37* | .15 | .31* | .34* | .33* | .35* |
| | Hostility | .40* | .45* | .25* | .25* | .37* | .36* | .34* | .42* | .34* | .34* | .38* | .33* | .49* | .60* | .36* | .33* | .35* | .49* | .34* | .48* |
| | Impulsivity | .25* | .19* | .23* | .22* | .36* | .34* | .31* | .35* | .30* | .32* | .30* | .31* | .39* | .49* | .31* | .32* | .29* | .40* | .30* | .43* |
| | Intimacy Avoidance | .36* | .41* | .14* | .18* | .26* | .03 | .23* | .14 | .22* | .01 | .19* | .11 | .18* | .14 | .22* | .05 | .23* | .15 | .27* | .21* |
| | Irresponsibility | .25* | .16* | .27* | .21* | .42* | .31* | .36* | .27* | .39* | .30* | .37* | .27* | .38* | .31* | .32* | .22* | .30* | .23* | .39* | .39* |
| | Manipulativeness | .42* | .34* | .16* | .01 | .22* | .11 | .18* | .12 | .19* | .14 | .25* | .17* | .27* | .37* | .26* | .13 | .26* | .20* | .26* | .24* |
| | Perceptual Dysregulation | .55* | .62* | .37* | .48* | .52* | .53* | .48* | .51* | .46* | .44* | .47* | .45* | .44* | .55* | .51* | .47* | .44* | .60* | .58* | .65* |
| | Perseveration | .57* | .58* | .38* | .32* | .60* | .62* | .52* | .54* | .54* | .52* | .50* | .44* | .45* | .44* | .46* | .41* | .47* | .50* | .49* | .52* |
| | Restricted Affectivity | .24* | .29* | .12 | .19* | .27* | .23* | .28* | .26* | .18* | .23* | .16* | .20* | .16* | .21* | .21* | .20* | .29* | .27* | .31* | .36* |
| | Rigid Perfectionism | .31* | .38* | .22* | .18* | .32* | .43* | .28* | .38* | .26* | .32* | .30* | .26* | .27* | .27* | .26* | .24* | .30* | .44* | .25* | .34* |
| | Risk Taking | .25* | .04 | .19* | .02 | .25* | -.02 | .22* | -.01 | .17* | -.07 | .18* | .01 | .23* | .18* | .24* | .01 | .23* | .15 | .24* | .12 |
| | Separation Insecurity | .53* | .48* | .38* | .26* | .46* | .47* | .48* | .40* | .50* | .51* | .50* | .43* | .43* | .33* | .46* | .45* | .41* | .32* | .50* | .34* |
| | Submissiveness | .35* | .28* | .23* | .15 | .37* | .30* | .35* | .34* | .34* | .29* | .33* | .21* | .18* | .13 | .26* | .16 | .27* | .21* | .30* | .24* |

*(Continued)*

**Table 7.** (*Continued*)

| SCL90R | GSI | | SOM | | O-C | | I-S | | DEP | | ANX | | HOS | | PHOB | | PAR | | PSY | |
|---|---|---|---|---|---|---|---|---|---|---|---|---|---|---|---|---|---|---|---|---|
| | non clin | clin | non clin | clin | non clin | clin | non clin | clin | non clin | clin | non clin | clin | non clin | Clin | non clin | clin | non clin | clin | non clin | clin |
| Suspiciousness | .39* | **.55***  | .25* | .36* | .37* | **.43*** | .37* | **.58*** | .34* | .38* | .31* | **.42*** | .27* | **.56*** | .30* | **.44*** | **.43*** | **.69*** | .36* | **.52*** |
| Unusual Beliefs | **.40*** | **.49*** | .32* | .39* | .38* | **.41*** | .32* | .37* | .32* | .29* | .30* | .36* | .38* | **.43*** | **.40*** | **.40*** | .37* | **.50*** | *.37** | **.44*** |
| Withdrawal | .37* | **.46*** | .25* | .29* | .36* | .33* | **.43*** | **.47*** | .28* | **.43*** | .27* | .38* | .26* | .28* | .36* | .34* | .35* | .39* | **.41*** | **.46*** |

Note. Nnonclinical = 471, Nclinical = 314.

\* = p < .005 (Bonferroni corrected level of significance), correlations > .40 are in bold, non-significant correlations are in italics, the corresponding PID-5 and SCL-90-R scales are marked in gray. GSI: Global Severity Index; SOM: Somatization; O-C: Obsessive-compulsive; I-S: Interpersonal sensitivity; DEP: Depression; ANX: Anxiety; HOS: Hostility; PHOB: Phobic anxiety; PAR: Paranoid ideation; PSY: Psychoticism.

has only a weak relationship with Avoidant- and Obsessive-Compulsive PD. Restricted Affectivity is weakly correlated to Schizotypal- and Obsessive-Compulsive PD and the correlation between Borderline PD and Risk Taking is also weak.

Total score of SCL-90-R has significant (Bonferroni corrected) correlations with all PID-5 scales, except for Risk Taking in the clinical sample. Correlations are moderate to strong, except for Attention Seeking, Callousness, Grandiosity, Impulsivity, Irresponsibility, Restricted Affectivity in both samples as well as Deceitfulness and Submissiveness in clinical sample. The effect sizes of these latter correlations are weak.

At domain level, the PID-5 negative affectivity, detachment, disinhibition and psychoticism show significant, moderate or strong correlations with all the SCL90-R scales. PID-5 antagonism has weak relationship to SCL-90 R scales.

With regards to the facet level, as expected, the strongest relations were found between the PID-5 and the SCL-90-R counterparts. PID-5 anxiousness shows moderate correlation with the SCL-90-R anxiety scale and the PID-5 depressivity displays high correlation with the SCL-90-R depression scale. Furthermore, PID-5 hostility is correlated with SCL-90-R hostility, while PID-5 suspiciousness correlates with the SCL-90-R paranoid ideation, all of them with moderate effect sizes.

However, there are several PID-5 traits that do not correlate with some of the SCL-90-R dimensions in the clinical sample, these are: callousness, deceitfulness, grandiosity, intimacy avoidance, manipulativeness, risk taking and submissiveness.

**3.2.2. Differences between clinical and non-clinical samples, gender, age and education.** Associations between PID-5 domains and facets comparing clinical and non-clinical groups, gender differences, as well as correlations with age and education are reported in Table 8.

*3.2.2.1. Clinical and non-clinical differences.* MANOVA was run to analyze clinical and non-clinical differences along with the 25 personality disorder traits and the 5 domains. Based on the 25 traits we found significant differences with a large effect size (Wilks's l = .449; $F_{(25, 759)}$ = 37.31; p < .001; $h^2$ = .551). More specifically, the Manipulativness, Deceitfulness and Risk Taking traits did not show significant differences between the clinical and non-clinical groups; for all the other traits, scores were significantly higher in the clinical group. At the domain level, the difference was also significant with a large effect size (Wilks's l = .564; $F_{(25, 779)}$ = 120.606; p < .001; $h^2$ = .436).

*3.2.2.2. Gender differences.* According to our results, the male and female profiles of personality disorder traits were significantly different and the difference had a large effect size in both the clinical (Wilks's l = .755; $F_{(25, 288)}$ = 3.735; p < .001; $h^2$ = .245) and the non-clinical

**Table 8. Descriptives of PID-5, differences between clinical and non-clinical samples, gender differences, correlation with age and educational level.**

| | Descriptive statistic of PID-5 scales | | | | | | Differences | | | Correlation | | | |
| --- | --- | --- | --- | --- | --- | --- | --- | --- | --- | --- | --- | --- | --- |
| | Total Healthy N = 588 | Male Healthy N = 251 | Female Healthy N = 337 | Total Clinical N = 314 | Male Clinical N = 95 | Female Clinical N = 219 | Clinical- Non clinical differences | Gender differences | | with age | | with educational level | |
| | M (SD) | M (SD) | M (SD) | M (SD) | M (SD) | M (SD) | $r_{M-W}$ | $r_{M-W}$ | $r_{M-W}$ | $r_p$ | $r_p$ | $r_s$ | $r_s$ |
| **Negative Affectivity** | .85 (.52) | .78 (.53) | .91 (.51) | 1.63 (.62) | 1.45 (.54) | 1.70 (.64) | .52** | .15** | .21** | -.17** | .01 | -.18** | -.15* |
| **Detachment** | .53 (.40) | .61 (.46) | .47 (.34) | 1.23 (.53) | 1.20 (.51) | 1.24 (.55) | .43** | .13** | .03 | .08 | .01 | .09* | -.10 |
| **Antagonism** | .54 (.45) | .66 (.48) | .44 (.39) | .60 (.51) | .73 (.53) | .54 (.49) | .48** | .26** | .19** | -.17** | -.26** | -.16** | -.02 |
| **Disinhibition** | .60 (.45) | .64 (.48) | .58 (.42) | 1.26 (.60) | 1.25 (.55) | 1.26 (.63) | .44** | .05 | .01 | -.17** | -.14* | -.17** | -.13* |
| **Psychoticism** | .45 (.45) | .53 (.49) | .40 (.42) | .85 (.61) | .78 (.54) | .88 (.64) | .30** | .14** | .06 | -.17** | -.12 | -.17** | -.12* |
| Anhedonia | .59 (.49) | .68 (.55) | .53 (.44) | 1.51 (.58) | 1.47 (.51) | 1.53 (.61) | .62** | .14** | .06 | -.03 | .07 | .01 | -.10 |
| Attention Seeking | .66 (.61) | .80 (.65) | .55 (.55) | .93 (.75) | 1.13 (.75) | .85 (.73) | .17** | .20** | .19** | -.18** | -.25** | -.18** | .07 |
| Anxiousness | .93 (.70) | .85 (.69) | .99 (.70) | 1.75 (.68) | 1.67 (.63) | 1.79 (.70) | .48** | .11** | .10 | -.17** | .06 | -.17** | -.13* |
| Callousness | .30 (.37) | .43 (.42) | .20 (.28) | .52 (.43) | .64 (.43) | .47 (.42) | .33** | .37** | .21** | -.06 | -.20** | -.08* | -.10 |
| Deceitfulness | .46 (.46) | .60 (.49) | .36 (.40) | .55 (.55) | .69 (.55) | .50 (.54) | .06 | .28** | .21** | -.19** | -.30** | -.16** | -.01 |
| Depressivity | .42 (.47) | .46 (.52) | .39 (.43) | 1.46 (.73) | 1.32 (.64) | 1.52 (.76) | .33** | .03 | .13* | -.14** | -.06 | -.12** | -.10 |
| Distractibility | .67 (.58) | .7 2 (.62) | .63 (.56) | 1.57 (.72) | 1.56 (.69) | 1.58 (.74) | .54** | .07 | .02 | -.17** | .04 | -.17** | -.14* |
| Eccentricity | .60 (.67) | .75 (.73) | .49 (.59) | 1.30 (.88) | 1.30 (.81) | 1.30 (.91) | .39** | .21** | .01 | -.23** | -.18** | -.25** | -.09 |
| Emotional Lability | 1.05 (.61) | .88 (.60) | 1.17 (.59) | 1.75 (.73) | 1.48 (.63) | 1.87 (.74) | .43** | .26** | .26** | -.07 | -.10 | -.08 | -.13* |
| Grandiosity | .45 (.53) | .60 (.59) | .34 (.45) | .53 (.60) | .67 (.66) | .46 (.56) | .04 | .25** | .16** | -.09* | -.09 | -.09* | .03 |
| Hostility | .81 (.54) | .87 (.55) | .77 (.53) | 1.19 (.69) | 1.25 (.67) | 1.16 (.70) | .27** | .09* | .07 | -.08* | -.24** | -.08* | -.14* |
| Impulsivity | .65 (.58) | .63 (.57) | .66 (.58) | 1.22 (.84) | 1.19 (.81) | 1.23 (.86) | .32** | .03 | .02 | -.07 | -.20** | -.07 | -.13* |
| Intimacy Avoidance | .44 (.46) | .50 (.50) | .40 (.43) | .93 (.75) | .86 (.70) | .96 (.77) | .33** | .09* | .05 | .09* | -.04 | .09* | -.07 |
| Irresponsibility | .49 (.44) | .56 (.48) | .44 (.40) | .98 (.65) | 1.05 (.61) | .95 (.66) | .38** | .12** | .08 | -.19** | -.20** | -.20** | -.01 |
| Manipulativeness | .69 (.58) | .79 (.62) | .62 (.54) | .72 (.67) | .84 (.68) | .67 (.67) | .01 | .14** | .13* | -.17** | -.27** | -.15** | -.02 |
| Perceptual Dysregulation | .31 (.40) | .34 (.43) | .28 (.37) | .68 (.57) | .54 (.49) | .75 (.59) | .37** | .07 | .17** | -.20** | -.11 | -.23** | -.14* |
| Perseveration | .76 (.55) | .79 (.58) | .74 (.53) | 1.50 (.62) | 1.43 (.53) | 1.53 (.66) | .50** | .03 | .07 | -.05 | -.01 | -.07 | -.10 |
| Restricted Affectivity | .74 (.58) | .94 (.62) | .60 (.51) | 1.08 (.61) | 1.17 (.64) | 1.03 (.60) | .27** | .29** | .12* | -.01 | -.01 | .01 | -.11 |
| Risk Taking | 1.14 (.57) | 1.31 (.56) | 1.02 (.54) | 1.11 (.61) | 1.11 (.58) | 1.11 (.62) | .04 | .26** | .01 | -.21** | -.21** | -.20** | -.09 |
| Rigid Perfectionism | .93 (.65) | .96 (.66) | .91 (.64) | 1.30 (.73) | 1.22 (.71) | 1.33 (.74) | .24** | .04 | .08 | -.01 | -.10 | -.01 | -.11 |
| Separation Insecurity | .58 (.58) | .59 (.58) | .57 (.58) | 1.36 (.84) | 1.20 (.79) | 1.43 (.85) | .44** | .03 | .12* | -.17** | .10 | -.18** | -.07 |
| Submissiveness | .90 (.66) | .88 (.66) | .91 (.66) | 1.42 (.81) | 1.29 (.67) | 1.48 (.86) | .30** | .02 | .11 | -.09 | .03 | -.01 | .05 |
| Suspiciousness | .85 (.45) | .88 (.50) | .82 (.41) | 1.16 (.60) | 1.14 (.58) | 1.17 (.62) | .25** | .04 | .03 | .03 | -.16** | .04 | -.29** |
| Unusual Beliefs | .45 (.50) | .48 (.50) | .43 (.49) | .57 (.62) | .52 (.52) | .59 (.65) | .07 | .07 | .02 | .01 | -.01 | .01 | -.15** |
| Withdrawal | .56 (.55) | .65 (.61) | .49 (.48) | 1.23 (.71) | 1.24 (.70) | 1.22 (.72) | .45** | .12** | .01 | .12** | .01 | .11** | -.09 |

Note

* p < .05

** p < .01

M = mean; SD = Standard Deviation; $r_{M-W}$: effect size of Mann-Whitney test, $r_p$ = Pearson correlation coefficient, $r_s$ = Spearman correlation coefficient.

sample (Wilks's l = .70; F (25, 445) = 7.64; p < .001; $h^2$ = .30). Significant difference between males and females was found at the domain level as well, and the difference had a large effect size in both the clinical (Wilks's l = .896; F (5, 308) = 7.14; p < .001; $h^2$ = .104) and the non-

clinical (Wilks's l = .857; F (5, 465) 15.564; p < .001; $h^2$ = .0143) sample. Along the higher order domains, in the non-clinical sample we found significant differences between women and men in every domain except for Disinhibition; females scored significantly higher in the Negative Affectivity domain and lower in the other domains. However, based on the effect sizes, difference in the Negative Affectivity domain is trivial, in Detachment and Psychoticism difference is small, and in Antagonism the difference is moderate. In the clinical group at the domain level we found significant differences only in the Negative Affectivity domain, where females scored higher, and in the Antagonism domain, where males scored higher. On the facet level, there are significant gender differences along 17 traits in the non-clinical group; effect sizes are between .011 and .12.

*3.2.2.3. Age and education.* There are several traits and domains showing significant correlations with age, but the correlation coefficients are very weak, except for the Deceitfulness trait and the Antagonism domain related to it, where we found a weak negative correlation, mostly in the clinical sample.

We also found some significant but very weak correlations with education on both the domain and the facet levels. There is a weak negative correlation between Suspiciousness and education in the clinical sample (r = 28).

## 4. Discussion

The results of the present study supported the hypothesis that the Hungarian translation of the PID-5 is a reliable and valid measure of the proposed DSM-5 maladaptive traits, and our findings provide valuable information for assessing personality pathology in non-clinical and clinical populations. Specifically, we found that

1. based on the Cronbach alphas, most of the scales have appropriate reliability in both samples, except for the Suspiciousness facet in the non-clinical sample; however, item-level CFA resulted in several poorly fitting trait scales

2. test-retest reliability estimates and effect sizes reflect small to no changes;

3. we did not find significant differences among the three different models in terms of the factor structure of PID-5 traits, however, the APA model was found to fit slightly better than the others

4. validity of the PID-5 instrument is acceptable in both the clinical and the non-clinical sample, as PID-5 scales show a significant correlations with the equivalent scales of SCID-II and SCL-90-R

5. there are significant differences between the personality disorder traits of clinical and non-clinical populations

6. male and female profiles of personality disorder traits were significantly different in both samples, but in the non-clinical sample there are more significant differences at the facet and the domain level as well;

7. there are significant negative correlations between age and PID-5 domains in the non-clinical sample, except for the Detachment domain; and in the clinical sample, Antagonism and Disinhibition is significantly and negatively correlated with age

8. we did not find strong significant associations between education and PID-5 traits and domains

Based on our results our conclusion is that we recommend the Hungarian version of the PID-5 for future clinical research and practice. In the following, we highlight selected findings that contribute to the current literature on the PID-5 and the new DSM-5 trait model in general.

First, according to the analysis of the internal consistency index (Cronbach alpha) and the item-level, CFA fit indexes of PID-5 facets, six scales of the Hungarian PID-5 were poorly fitting based on several fit indices in both samples. These are Emotional Lability, Unusual Beliefs, Depressivity, Perseverance, Risk Taking, and Restricted Affectivity. Some of the fit indices of Emotional Lability and Perseverance did not meet minimal standards in other studies either [18]. Restricted Affectivity and Risk Taking also showed inadequate model fit in another research analyzing the psychometric properties of another translation of the Hungarian PID-5 [3, 19]. In our research, the only scale with an unacceptable Cronbach alpha level was Suspiciousness (.54) in the non-clinical sample. The reliability of the Suspiciousness scale was weak in several other psychometric analysis of PID-5 [3], suggesting that this is not a specific characteristic of the Hungarian version of PID-5. We analyzed the factor loadings of the items of the 6 problematic scales and some of them were very low, under .2 (Emotional Lability: item 18, 138, 181, Unusual Beliefs: item 143, Depressivity: item 86, Perseverance: item 46, Restricted Affectivity: item 45, Risk taking: items 98, 164). It might be necessary to revise and rewrite these problematic items in order to increase reliability.

Second, temporal consistency of the Hungarian PID-5 has been confirmed by a one-month test-retest reliability analysis. Based on Cohen's d values, small to no change was observed along with PID-5 facet scales within the one-month interval. The small amount of change observed in the Negative Affectivity (d = −.30), and in the Disinhibition (d = −.20) domains is similar to the findings of Wright and colleagues [8] who observed a small degree change in 6 out of the 25 traits after 1,5 years. Wright considers that the small negative change might be explained by some amelioration of symptomatology and shows that although PD traits are highly stable, some traits might be dynamically associated with the change of psychosocial functioning over time. In our study, test-retest correlation coefficients were between .53 (Callousness) and .86 (Risk-taking), which reflects a moderate to strong connection between the two measurements. In comparison with other self-report measures of psychopathology in other studies [20], PID-5 scales have similar stability coefficients.

Third, we did not find significant differences among the three different models in terms of the factor structure of PID-5 traits. According to our analysis, those models that include all the 25 traits fit weakly in every domain based on the CFI scores. However, the APA model—that builds up the five domains out of 15 facets—, was found to fit slightly better than the others. The Disinhibition domain proved to be a good fit in both samples, and the Detachment domain was fitting well in the clinical sample based on every fit index. The weak CFI scores could be explained by that many items of the scales load onto several factors simultaneously, however, this does not mean that the scales are inconsistent because they have acceptable Cronbach alpha levels.

Fourth, validity of the PID-5 instrument is acceptable in the clinical and the non-clinical sample as well. PID-5 scales are significantly correlated with SCID-II total PD score and SCL-90-R global severity index. Furthermore, different personality disorder scores of SCID-II-PQ are significantly correlated with the expected PID-5 domains and traits with moderate or strong effect size. Moreover, PID-5 domains show significant correlations with all the SCL90-R scales, and as expected we found strong relations between the PID-5 facets and their SCL-90-R counterparts. These results support the hypotheses that the Hungarian PID-5 is a valid measurement of personality pathology.

However we found some unexpected results as well. PID-5 Intimacy avoidance shows only a weak, almost negligible relationship with Avoidant PD, and has no significant correlations with 7 out of the 9 SCL-90-R dimensions, but only in the clinical sample. According to these results, further studies are needed to understand how PID-5 Intimacy avoidance scale can describe the pathological avoidance behavior. We also found that PID-5 Risk taking scale does not correlate significantly with Global Severity Index of SCL-90-R, and 8 of the 9 SCL-90-R subscales in the clinical sample, and although its relationship is significant with Borderline PD, it is only with weak effect size. Furthermore, Risk Taking is one of the four PID-5 traits that does not differentiate significantly between clinical and non-clinical samples. Our results support that Risk taking is a reliable scale of PID-5, but the question arises as to how pathological it can be considered. There are two other PID-5 traits, Deceitfulness and Manipulativeness, which are not correlated significantly with many SCL-90-R dimensions. It suggests that these traits might be independent from psychological distress.

Fifth, we found significant differences among clinical and non-clinical participants in four domains (no difference in Antagonism) and in 22 out of 25 facets, with mostly medium effect sizes. (See Table 8.) In the majority of comparisons, clinical participants received significantly higher scores than non-clinical participants. No significant differences were observed for the facets of Deceitfulness, Manipulativeness and Risk Taking between the two samples; these facets all belong to the Antagonism domain. This finding is only partially in line with Bach and colleagues' findings [21], where they observed no significant differences between clinical and non-clinical groups for the facets of Manipulativeness and Attention Seeking. Scores in the Antagonism domain are relatively low in both samples, meaning that antisocial personality functioning was not characteristic in our clinical sample either. The low level of Antagonism domain in the clinical sample might be explained by the fact that our clinical sample consisted of the participants of the four weeks long psychotherapy program of our department and severe antisocial traits and the diagnosis of antisocial personality disorder with traits of primary psychopathy are exclusion criteria for our treatment program.

Sixth, we found significant differences between the profiles of females and males in PID-5 domains and facets in both samples. Females presented higher scores in the Negative Affectivity domain in both samples, while men score significantly higher in the Antagonism domain in both samples. However, based on effect sizes, the difference between genders in Negative Affectivity is small. In this domain, Emotional Lability scores are higher in females and the difference has medium effect size, but with Anxiousness and Separation Anxiety differences are negligible.

These results are partially in line with the findings of Bastiaens et al. [22] on a clinical sample, where females scored higher on Negative Affectivity and lower on Antagonism than males. However, Bastiaens and colleagues found that females scored higher on Disinhibition, which domain showed no difference in neither of our samples. Moreover, in our non-clinical sample at the facet level males scored higher on the Irresponsibility, Distractibility, Manipulativeness and Risk Taking scales, which is in accordance with the results of Fossati et al. [23], and with reports of higher scores on Conscientiousness in females [24, 25].

Seventh, regarding age, a significant negative relation was found with Negative Affectivity, Antagonism, Disinhibition, and Psychoticism in the non-clinical sample. In the clinical sample, only Antagonism and Disinhibition showed significant negative relation with age. In previous PID-5 studies, inverted relations were found between age and the facets of Antagonism, Disinhibition and Psychoticism [23]. In a different study, similar associations were found in a clinical sample [22]. In a non-clinical sample, again, an inverted relation was found between age and Antagonism and Disinhibition [26]. Our results are in line with the results of longitudinal studies in the US and cross-sectional studies from many cultures that were using self-

report measures of personality traits, namely that the broad factors of Neuroticism, Extraversion, and Openness to Experience decline from adolescence to adulthood, whereas Agreeableness and Conscientiousness increase [27].

Eighth, we did not find strong significant associations between education and PID-5 traits and domains. This is contradicted by some earlier findings, for example the study of Bastiaens et al. [22] found that Disinhibition was significantly negatively related to education.

Our results are limited by that our measure of personality disorders (SCID-II-PQ) was based only on self-reports. We did not measure the symptoms of Antisocial personality disorder since our ethical committee did not allow us to use the antisocial modul of the SCID-II-PQ for legal and ethical reasons. There was many data missing in the clinical sample, which led to a smaller sample size than originally planned.

Even in light of these limitations, our study shed new light on some aspects of PID5, as well as replicating some previous results [12, 28] in a linguistically and culturally different non-clinical and clinical sample.

## Author Contributions

**Conceptualization:** Zita S. Nagy, Bettina Bajzát, Bálint Hajduska-Dér, Zsolt Szabolcs Unoka.

**Data curation:** Zita S. Nagy, Bettina Bajzát, Zsolt Szabolcs Unoka.

**Formal analysis:** Zita S. Nagy.

**Funding acquisition:** Zsolt Szabolcs Unoka.

**Investigation:** Bettina Bajzát, Bálint Hajduska-Dér, Zsolt Szabolcs Unoka.

**Methodology:** Zita S. Nagy, Bettina Bajzát, Zsolt Szabolcs Unoka.

**Project administration:** Bettina Bajzát.

**Supervision:** Zsolt Szabolcs Unoka.

**Validation:** Zsolt Szabolcs Unoka.

**Writing – original draft:** Zita S. Nagy, Ella Salgó, Bettina Bajzát, Zsolt Szabolcs Unoka.

**Writing – review & editing:** Zita S. Nagy, Bálint Hajduska-Dér, Zsolt Szabolcs Unoka.

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
