## [Decision Letter · Decision Letter 0]

12 Jan 2022

PONE-D-21-36847Reliability and validity of the Hungarian Version of the Personality Inventory for DSM-5 (PID-5)PLOS ONE

Dear Dr. Unoka,

Thank you for submitting your manuscript to PLOS ONE. After careful consideration, we feel that it has merit but does not fully meet PLOS ONE’s publication criteria as it currently stands. Therefore, we invite you to submit a revised version of the manuscript that addresses the points raised during the review process.

We look forward to receiving your revised manuscript.

Kind regards,

Stephan Doering, M.D.

Academic Editor

PLOS ONE

“This work was supported by the

Hungarian National Research, Development and

Innovation Fund [grant numbers NKFI-132546, NKFI-129195]. Zsolt Unoka received these grants.”

5. Please include a copy of Table 1 which you refer to in your text on page 4.

Reviewers' comments:

Reviewer's Responses to Questions

**Comments to the Author**

1. Is the manuscript technically sound, and do the data support the conclusions?

Reviewer #1: Yes

Reviewer #2: Yes

2. Has the statistical analysis been performed appropriately and rigorously? 

Reviewer #1: Yes

Reviewer #2: Yes

3. Have the authors made all data underlying the findings in their manuscript fully available?

Reviewer #1: Yes

Reviewer #2: Yes

4. Is the manuscript presented in an intelligible fashion and written in standard English?

Reviewer #1: No

Reviewer #2: Yes

5. Review Comments to the Author

Reviewer #1: The paper presents the results of the study testing psychometric properties of the Hungarian version of the Personality Inventory for DSM-5 (PID-5). I think we have thorough, high-quality work here in terms of applied procedures, methodological issues and statistical analyses. What is more, the study have a potential to essentially contribute to the literature, particularly due to CFA examination of three different models of the PID-5 internal structure (i.e., relationships between facets and domains) as well as test-retest examination of reliability. Also an overview of FA methods used in previous studies analyzing PID-5 structure in different languages is a strength of the paper (although I had very hard time trying to find the Table 1, which presents this “mini-review”).

However, the manuscript in its current shape has many faults which definitely should be eliminated before publication. Below, I’m pointing out four such major concerns and several minor ones.

1. There is an another, published study on Hungarian adaptation of the PID-5 using another adaptation of the measure (Labancz et al., 2020). The Authors admitted that and referred to this publication, but just in the Discussion section. Meanwhile, it should be one of the main issues in the Introduction. The Authors should refer to the previous study pointing out its strengths and limitations, both psychometric and scientific ones. There are many advantages of the current research in relation to Labancz et al.’ study that should be mentioned (e.g., APA permission, test-retest reliability, validity criteria, CFA analyses) and convince the reader that the current study is an extension of the previous one (and propose adaptation of the PID-5 could be considered competitive to or even better than the previous one).

2. The considerably large clinical sample is one of the largest strength of the study. However, the reader has no chance to learn about its characteristics. There are some additional information but just in the Discussion section, which is a definitely too late. Moreover, the equivalence of the clinic and non-clinic samples in term of gender and age is impressive, but still more information concerning the procedure of achieving that is needed. I would also be grateful for some more information about how it was happened that n = 314 became n = 222 in the complete factor structure analysis (as it is an essential loss of the sample size). And what does it exactly mean, that “In all the other type of analysis, missing data were left out and the clinical sample size was 314”? I think this issue should be mentioned as a limitation of the study. Summing up, the Participants and procedure section should be enriched and improved.

3. As mentioned, I found the CFA test of the three different models of the PID-5 internal structure (i.e., by APA, early Krueger, and Watters and Bagby) is the strength of the study. However, these models should be definitely more precisely described (and maybe illustrated?), in particular with the explanation of differences between the first and the third one.

4. The last major remark is the most formal and least specific one. In my opinion the paper suffer from many editing errors and generally rather poor language. There are also many inconsistences, lacks of information or severe lacks of precision in some parts of the manuscript (please, see below), which make them “messy”.

Therefore, I strongly recommend professional English proofreading and edition of the manuscript before resubmission. Moreover, in my opinion the Introduction section should be better structured, at least by division of the text into more paragraphs. The Authors should also check the numbers of Tables (e.g., Table 3 on p. 9 and 14) and screen the whole manuscript for typographical errors and inconsistent information. Some examples and another remarks are given below:

- p.2: “PDNOS”, “APA”, “DSM” – the abbreviations should be explained while the first use;

- p.2: “Since its publication, more than hundred publications investigated the validity of the AMDP in different languages” – some relevant reference should be added;

- p.3 “APA (….) provided different kinds of measures (self-report, informant, rating scale) for assessing maladaptive personality traits. The self-report version is the Personality Inventory for the DSM–5 (PID–5)” – imprecise statement as PID-5 is also other-informant measure;

- p.3: inconsistent expression. Firstly the Authors claim: “Internal consistencies of PID-5 domains and facets have been studied extensively, and strong internal consistency scores were demonstrated.”, a and in the next sentence: “Coefficient alphas (…) for facets ranged from .46-.77 to .94-.96, demonstrating some possible issues with scale reliability at the facet level”;

- p.3: “In order to eliminate cross-loadings of facets, three facets with the highest facet to domain factor loadings (…)” – please, rephrase;

- p.4 and others: the Authors use the term “healthy” in reference to the none-clinical sample. It seems that the term “non-clinical” is more appropriate as we know a little about the health level of the community sample;

- p.6: “Twelve PDs were assessed; all cluster A, B, and C PDs and Passive-Aggressive and Depressive PD”. How is that possible since items referring to antisocial personality disorder were left out and only 104 items were used? How this information relates to the use of just a total score of SCID-2 in analyses? Wouldn’t be the SCID-2 specific disorders scores valuable as validity criteria for the PID-5 scales?

- p.7: “A series of confirmatory factor analyses (CFA) was conducted to test the unidimensionality of the PID-5 scales”. It is a very frugal, unsatisfactory and imprecise description of the CFA procedure used in the study;

- p.8: “In summary, most scales have appropriate reliability, except for the Suspiciousness facet, the reliability of which is low” – it should be added that this conclusion concerns only the non-clinical sample. The same problem appears on p. 12 (point 1);

- p.9: “SCID-II total PD scores are significant in both the clinical and the non-clinical groups even after Bonferroni correction.” – please, rephrase;

- p.10: “Associations between PID-5 domains and facets between clinical and non-clinical groups, gender, age and education are reported in Table 6” – please, rephrase;

- p.11: the direction of significant gender differences should be given to enhance the readability.

Reviewer #2: Beyond the Hungarian translation of the PID5, the authors propose interesting results on the relationship between AMPD and clinical disorders, using the SCL 90r. It would have been useful if elements could have been mentioned about the relations between the subscales of this questionnaire and the data of the PID 5. But this does not penalize the quality of the present work and could obviously be the subject of a later work allowing to deepen the relations between personality disorders and clinical disorders.

It is regrettable that the authors do not mention the conditions of the translation of the PID 5 into Hungarian. Was a return translation made? Is this translation freely available so that Hungarian speakers can use it?

Finally, an important result appears in this work that is not sufficiently discussed. This is the lack of stability at 1 month in Negative Affectivity and Disinhibition. Pathological personality traits, like personality disorders, are known to have stable and long-lasting modalities. How, could the authors explain their results regard to the literature.

6. PLOS authors have the option to publish the peer review history of their article (what does this mean?). If published, this will include your full peer review and any attached files.

Reviewer #1: **Yes: **Włodzimierz Strus

Reviewer #2: **Yes: **Serge Combaluzier

---

## [Author Response · Author response to Decision Letter 0]

9 Mar 2022

February 17, 2022

Stephan Doering, M.D.

Academic Editor

PLOS ONE

Manuscript Number: PONE-D-21-36847

Title: Reliability and validity of the Hungarian Version of the Personality Inventory for DSM-5 (PID-5)

Dear Professor Stephan Doering,

Thank you very much for your letter, which provided us with the opportunity to revise our manuscript.

Based on the helpful suggestions of the reviewers, we have revised the manuscript carefully. I have enclosed a revised version of the above paper for submission to PLOS ONE. We have addressed the comments raised by the reviewers.

Point-by-point responses to the reviewers’ comments are listed as follows.

- Reviewer 1. Comment 1.

“There is an another, published study on Hungarian adaptation of the PID-5 using another adaptation of the measure (Labancz et al., 2020). The Authors admitted that and referred to this publication, but just in the Discussion section. Meanwhile, it should be one of the main issues in the Introduction. The Authors should refer to the previous study pointing out its strengths and limitations, both psychometric and scientific ones. There are many advantages of the current research in relation to Labancz et al.’ study that should be mentioned (e.g., APA permission, test-retest reliability, validity criteria, CFA analyses) and convince the reader that the current study is an extension of the previous one (and propose adaptation of the PID-5 could be considered competitive to or even better than the previous one).”

Response 1.

We thank the Reviewer for this comment. We included the following paragraph in the introduction:

“Although another study has already been published about the psychometric properties of the PID-5 in a Hungarian clinical and community sample [19], we found that this study had some psychometric and scientific limitations besides its strengths, and it is worth using another adaptation of the PID-5 mesure. First of all, our Hungarian translation of the PID-5 scale had been approved by the American Psychiatric Association (APA). Moreover, we added some important extensions to the investigation of the psychometric properties of the PID-5 by including a 1-month test-retest reliability of the scales, by using SCID-II and SCL-90-R correlations as validity criteria, and by presenting three different models of the PID-5 internal structure in the confirmatory factor analysis.” 

- Reviewer 1. Comment 2.

“The considerably large clinical sample is one of the largest strength of the study. However, the reader has no chance to learn about its characteristics. There are some additional information but just in the Discussion section, which is a definitely too late. Moreover, the equivalence of the clinic and non-clinic samples in term of gender and age is impressive, but still more information concerning the procedure of achieving that is needed. I would also be grateful for some more information about how it was happened that n = 314 became n = 222 in the complete factor structure analysis (as it is an essential loss of the sample size). And what does it exactly mean, that “In all the other type of analysis, missing data were left out and the clinical sample size was 314”? I think this issue should be mentioned as a limitation of the study. Summing up, the Participants and procedure section should be enriched and improved.”

Response 2.

We thank the Reviewer for this comment. We added the following two paragraphs to the Method section:

“Based on SCID II interview, 173 participants (55,1 %) in the clinical sample had at least one diagnosis of personality disorder, 117 subjects (37,3%) had multiple diagnoses (two or more personality disorders). No information was available on the clinical history of 64 individuals. The distribution of the different types of personality disorder was as follows:

Avoidant 21,6% (54 subjects); Dependent 16,06% (40 subjects); Obsessive-compulsive 18,145% (45 subjects); Passive-aggressive 13,2% (33 subjects); Depressive 24,4% (61 subjects); Paranoid 14,8% (37 subjects); Schizotype 2,4% (6 subjects); Schizoid 1,2% (3 subjects); Histrionic 14% (35 subjects); Narcissistic 17,6% (44 subjects) and Borderline 30,8% (77 subjects).”

“In the clinical sample some of the data was missing, and these missing data were not replaced, therefore the sample sizes vary in each analysis. There were 92 individuals who did not mark an answer for at least one item of the PID-5 questionnaire. One participant was excluded from the analysis due to too much missing data (50%), for the others, the number of missing data did not exceed 10%. The sample size was the lowest (222) in the analyzes where all PID-5 items were included in the CFA. In CFAs when the complete factor structure was analyzed, the sample sizes varied between 222-241. Sample sizes ranged from 295 to 307 for item-level analyzes and from 265 to 277 for domain-level analyzes. No data were missing in the analyzes on facet- and domain scales, as the scores on the scales were calculated by averaging.”

- Reviewer 1. Comment 3.

“As mentioned, I found the CFA test of the three different models of the PID-5 internal structure (i.e., by APA, early Krueger, and Watters and Bagby) is the strength of the study. However, these models should be definitely more precisely described (and maybe illustrated?), in particular with the explanation of differences between the first and the third one.”

Response 3.

We thank the Reviewer for this comment. We expanded Table 1 in order to present the differences between the three models more precisely. 

- Reviewer 1. Comment 4.

“The last major remark is the most formal and least specific one. In my opinion the paper suffer from many editing errors and generally rather poor language. There are also many inconsistences, lacks of information or severe lacks of precision in some parts of the manuscript (please, see below), which make them “messy”.

Therefore, I strongly recommend professional English proofreading and edition of the manuscript before resubmission. Moreover, in my opinion the Introduction section should be better structured, at least by division of the text into more paragraphs. The Authors should also check the numbers of Tables (e.g., Table 3 on p. 9 and 14) and screen the whole manuscript for typographical errors and inconsistent information. Some examples and another remarks are given below:

- p.2: “PDNOS”, “APA”, “DSM” – the abbreviations should be explained while the first use;

- p.2: “Since its publication, more than hundred publications investigated the validity of the AMDP in different languages” – some relevant reference should be added;

- p.3 “APA (….) provided different kinds of measures (self-report, informant, rating scale) for assessing maladaptive personality traits. The self-report version is the Personality Inventory for the DSM–5 (PID–5)” – imprecise statement as PID-5 is also other-informant measure;”

Response 4.

We thank the Reviewer for this comment. We corrected the above mistakes and included the required references.

We also had the article proofread, checked the number of the Tables, corrected the typos and divided the Introduction into more paragraphs. 

- Reviewer 1. Comment 5.

“- p.3: inconsistent expression. Firstly the Authors claim: “Internal consistencies of PID-5 domains and facets have been studied extensively, and strong internal consistency scores were demonstrated.”, a and in the next sentence: “Coefficient alphas (…) for facets ranged from .46-.77 to .94-.96, demonstrating some possible issues with scale reliability at the facet level”;

- p.3: “In order to eliminate cross-loadings of facets, three facets with the highest facet to domain factor loadings (…)” – please, rephrase”

Response 5.

We thank the Reviewer for this comment. We changed the quoted sentences to the following: 

“Internal consistencies of PID-5 domains and facets have been studied extensively, and mixed internal consistency scores were demonstrated.”

“In order to eliminate cross-loadings of facets, three traits with the highest factor loadings were selected for each domain.”

- Reviewer 1. Comment 6.

“p.4 and others: the Authors use the term “healthy” in reference to the non-clinical sample. It seems that the term “non-clinical” is more appropriate as we know a little about the health level of the community sample”

Response 6.

We thank the Reviewer for this comment. We changed “healthy” to “non-clinical” everywhere in the text. 

- Reviewer 1. Comment 7.

“p.6: “Twelve PDs were assessed; all cluster A, B, and C PDs and Passive-Aggressive and Depressive PD”. How is that possible since items referring to antisocial personality disorder were left out and only 104 items were used? How this information relates to the use of just a total score of SCID-2 in analyses? Wouldn’t be the SCID-2 specific disorders scores valuable as validity criteria for the PID-5 scales?”

 Response 7.We thank the Reviewer for this comment. Following your question we corrected the description of the analysis concerning the relationship between PID-5 and SCID-II scales. Our original decision to use the total score only was part of the effort to make the manuscript concise, but based on your comment, in the revised manuscript we calculated the PID-5 scales’ relationship with those personality disorders that were part of the DSM-5 hybrid personality disorder model, since this is what the PID-5 is designed to measure. However, we did not measure the symptoms of Antisocial personality disorder since our ethical committee did not allow us to use the antisocial modul of the SCID-II-PQ for legal and ethical reasons. 

- Reviewer 1. Comment 8.

“- p.7: “A series of confirmatory factor analyses (CFA) was conducted to test the unidimensionality of the PID-5 scales”. It is a very frugal, unsatisfactory and imprecise description of the CFA procedure used in the study;”

Response 8.

We thank the Reviewer for this comment. We included the following paragraph under 2.3 to explain the procedure:

“Confirmatory factor analysis (CFA) was conducted to test the unidimensionality of the PID-5 scales and to explore the factor structure of PID-5 facets and domains. Model fit was evaluated using the usual fit indexes: Comparative Fit Index (CFI), Root Mean Square Error of Approximation (RMSEA), Standardized Root Mean Square Residual (SRMR). The CFI is acceptable if the value is higher than .90, the RMSEA and SRMR indexes should be less than .10 for adequate fit [15,16,17].”

- Reviewer 1. Comment 9.

“p.8: “In summary, most scales have appropriate reliability, except for the Suspiciousness facet, the reliability of which is low” – it should be added that this conclusion concerns only the non-clinical sample. The same problem appears on p. 12 (point 1);”

Response 9.

We thank the Reviewer for this comment, we added this precision to the sentence.

- Reviewer 1. Comment 10.

“- p.9: “SCID-II total PD scores are significant in both the clinical and the non-clinical groups even after Bonferroni correction.” – please, rephrase;

- p.10: “Associations between PID-5 domains and facets between clinical and non-clinical groups, gender, age and education are reported in Table 6” – please, rephrase;

- p.11: the direction of significant gender differences should be given to enhance the readability.”

Response 10.

We thank the Reviewer for this comment. We added the directions of gender differences and rephrased the quoted sentences to the following:

“We found that correlations of PID-5 domains and SCID-II total PD scores are significant in both the clinical and the non-clinical groups with a Bonferroni adjusted p-value.”

“Associations between PID-5 domains and facets comparing clinical and non-clinical groups, gender differences, as well as correlations with age and education are reported in Table 6.”

- Reviewer 2. Comment 1.

“Beyond the Hungarian translation of the PID5, the authors propose interesting results on the relationship between AMPD and clinical disorders, using the SCL 90r. It would have been useful if elements could have been mentioned about the relations between the subscales of this questionnaire and the data of the PID 5. But this does not penalize the quality of the present work and could obviously be the subject of a later work allowing to deepen the relations between personality disorders and clinical disorders.”

Response 1.

We thank the Reviewer for this comment. We added a section and new tables to present the relations between the SCL-90-R and PID-5 under 3.2.1.:

“Correlations between PID-5 traits and SCID-II personality disorder scores and SCL-90-R scores are reported in Table 7 and 8. We found that correlations of PID-5 domains and SCID-II total PD scores are significant in both the clinical and the non-clinical groups with a Bonferroni adjusted p-value. Effect sizes are moderate or strong. Correlations between the traits and SCID-II scores are stronger in the clinical sample, where correlations are moderate to strong, except for weak correlations with Intimacy Avoidance, Submissiveness and Risk Taking. In the non-clinical sample, correlations between the traits and SCID-II are mostly moderate, with the exceptions of avoidance, Submissiveness, Risk Taking and Rigid Perfectionism, Restricted Affectivity and Attention Seeking, which are also weakly correlated. “

- Reviewer 2. Comment 2.

“It is regrettable that the authors do not mention the conditions of the translation of the PID 5 into Hungarian. Was a return translation made? Is this translation freely available so that Hungarian speakers can use it?”

Response 2.

We thank the Reviewer for this comment. You can find in the method part the detailed description of our translation.

"Two independent translators translated the PID-5 into Hungarian, then a group of experts compared the two translations and agreed on a first version. We distributed the first version among patients of the Department of Psychotherapy at Semmelweis University, who commented on it, and based on their comments, we made corrections to the translation. There was no back translation made."

- Reviewer 2. Comment 3.

“Finally, an important result appears in this work that is not sufficiently discussed. This is the lack of stability at 1 month in Negative Affectivity and Disinhibition. Pathological personality traits, like personality disorders, are known to have stable and long-lasting modalities. How, could the authors explain their results regard to the literature.”

Response 3.

We thank the Reviewer for this comment. We added the following explanation in the Discussion section:

“The small amount of change observed in the Negative Affectivity (d = –.30), and in the Disinhibition (d = –.20) domains is similar to the findings of Wright and colleagues [29] who observed a small degree change in 6 out of the 25 traits after 1,5 years. Wright considers that the small negative change might be explained by some amelioration of symptomatology and shows that although PD traits are highly stable, some traits might be dynamically associated with the change of psychosocial functioning over time.” 

Sincerely,

Dr. Zsolt Unoka

Professor, Department of Psychiatry and Psychotherapy, Semmelweis University, Budapest, Hungary

Address: 1083 Budapest, Balassa utca 6., Hungary

Tel./Fax: +36706158366

Email: unoka.zsolt@med.semmelweis-univ.hu

---

## [Editor Report · Decision Letter 1]

16 Mar 2022

Reliability and validity of the Hungarian Version of the Personality Inventory for DSM-5 (PID-5)

PONE-D-21-36847R1

Dear Dr. Unoka,

We’re pleased to inform you that your manuscript has been judged scientifically suitable for publication and will be formally accepted for publication once it meets all outstanding technical requirements.

Kind regards,

Stephan Doering, M.D.

Academic Editor

PLOS ONE

---

## [Editor Report · Acceptance letter]

5 Apr 2022

PONE-D-21-36847R1 

Reliability and validity of the Hungarian version of the Personality Inventory for DSM-5 (PID-5) 

Dear Dr. Unoka:

I'm pleased to inform you that your manuscript has been deemed suitable for publication in PLOS ONE. Congratulations! Your manuscript is now with our production department. 

Kind regards, 

on behalf of

Professor Stephan Doering 

Academic Editor

PLOS ONE